# Realizing nearly-zero dark current and ultrahigh signal-to-noise ratio perovskite X-ray detector and image array by dark-current-shunting strategy

Peng Jin [1,7], Yingjie Tang [2,3,7], Dingwei Li[2,3], Yan Wang[2,3], Peng Ran[1], Chuanyu Zhou[1], Ye Yuan[4], Wenjuan Zhu[1,5], Tianyu Liu[1], Kun Liang[2,3], Cuifang Kuang [1,6], Xu Liu [1], Bowen Zhu [2] ✉ & Yang (Michael) Yang [1,6] ✉

Although perovskite X-ray detectors have revealed promising properties, their dark currents are usually hundreds of times larger than the practical requirements. Here, we report a detector architecture with a unique shunting electrode working as a blanking unit to suppress dark current, and it theoretically can be reduced to zero. We experimentally fabricate the dark-current-shunting X-ray detector, which exhibits a record-low dark current of 51.1 fA at 5 V mm$^{-1}$, a detection limit of 7.84 nGy$_{air}$ s$^{-1}$, and a sensitivity of $1.3 \times 10^4$ μC Gy$_{air}^{-1}$ cm$^{-2}$. The signal-to-noise ratio of our polycrystalline perovskite-based detector is even outperforming many previously reported state-of-the-art single crystal-based X-ray detectors by serval orders of magnitude. Finally, the proof-of-concept X-ray imaging of a 64 × 64 pixels dark-current-shunting detector array is successfully demonstrated. This work provides a device strategy to fundamentally reduce dark current and enhance the signal-to-noise ratio of X-ray detectors and photodetectors in general.

Metal halide perovskites (MHPs) have demonstrated highly desirable properties including large mobility-lifetime product, strong X-ray absorption, and easy synthesis, showing sensitive X-ray detection performance[1–21]. Compared with the most prevalent amorphous Selenium (α-Se) detectors, MHPs possess extraordinary sensitivity to X-ray and high stopping power to ionization radiation[22–25]. Low-temperature solution process fabrication enables MHPs the easy integration with Si-based application-specific integrated circuits (ASIC) for signal readout[17]. Currently, the most prevalent direct X-ray detection application is photoconductive-type flat-panel X-ray detectors[26,27], they

generally require dark current densities well below 1 nA cm$^{-2}$ to maintain high detective quantum efficiency and dynamic range[28]. High sensitivity and low detection limit are also the key requirements to reduce the X-ray doses used on patients[3]. However, the high dark current of perovskite photoconduction detectors largely blocks their utilization in practical X-ray imagers[29]. First, a high dark current can quickly fill up the storage capacitance of TFT or CMOS pixels prior to X-ray illumination, which deteriorates the image contrast and dynamic range. Also, a large dark current significantly increases the shot noise and results in a poor signal-to-noise ratio (SNR)[30–32]. Lots of attention

[1]State Key Laboratory of Modern Optical Instrumentation, College of Optical Science and Engineering, Zhejiang University, Hangzhou 310007 Zhejiang, China. [2]Key Laboratory of 3D Micro/Nano Fabrication and Characterization of Zhejiang Province, School of Engineering, Westlake University, Hangzhou 310024 Zhejiang, China. [3]College of Information Science and Electronic Engineering, Zhejiang University, Hangzhou 310007 Zhejiang, China. [4]State Key Laboratory of Advanced Technology for Materials Synthesis and Processing, Wuhan University of Technology, Wuhan 430070, PR China. [5]College of Electronic and Optical Engineering, and College of Flexible Electronics (Future Technology), Nanjing University of Posts and Telecommunications, Nanjing 210023, P. R. China. [6]Intelligent Optics & Photonics Research Center Jiaxing Institute of Zhejiang University, Jiaxing Zhejiang 314041, China. [7]These authors contributed equally: Peng Jin, Yingjie Tang. ✉e-mail: zhubowen@westlake.edu.cn; yangyang15@zju.edu.cn

have been paid to reducing the dark current very recently, including 3D/2D perovskite heterojunction, dopant compensation of perovskite single crystals, and inserting insulating polymer between perovskite and electrode[33–39]. Given those encouraging progresses, the dark current density of perovskite X-ray detectors is still quite large, along with the notorious current drifting issue, becoming the biggest challenge toward its real-world applications.

In conventional photoconduction devices (Fig. 1a), the dark current and photocurrent are conducted in the same path and collected by the same electrodes, reducing the dark current is equivalent to increasing the resistance, which in turn also largely cuts the X-ray photocurrent. Herein, we propose a novel and general device structure, which can split the dark and photocurrent. It allows a fundamental and complete suppression of dark current (Fig. 1b). In this device structure, the electrons under dark are emitted from the source electrode and most of them can be collected by the top dark-current-shunting (DCS) electrode, or at least they will no longer be received by the drain, which gives the zero source-drain dark current. Under X-rays (Fig. 1c), the excited electrons, are first generated from X-ray-sensitive materials and transported to a lateral conduction channel with higher carrier mobility, thus, the captured carriers can transport rapidly in the conduction channel, reinject and recirculate swiftly between source and drain contacts, due to the long lifetime of the conduction channel, the captured carriers can transport longer before the recombination[40]. The amount of charges passing through the cross section of a conductor per unit of time is much greater. Therefore, these carriers in the conduction channel can offer much stronger photocurrent than the case with the mere X-ray sensitive materials[39–41]. Even some of the X-ray induced electrons will be attracted by DCS electrode, with the high photoconductive gain effect of the conduction channel, the photocurrent is still quite strong. Then the signal current is collected by the drain under an electric field applied from drain to source. In this scenario, the drain only receives X-ray-generated electrons, simultaneously gaining nearly-zero dark current and relatively high X-ray photocurrent.

In this study, we proposed and experimentally demonstrated the DCS detector that can effectively shunt the dark current, without much weakening of the photocurrent. The mobile electrons under dark can theoretically be fully depleted. The experimentally measured dark current is only 51.1 fA with no observable baseline drift, approaching the lowest measurable current of a typical semiconductor analyzer. As a result, the signal-to-noise ratio (SNR) was improved by two orders of magnitude. In addition to the high SNR single-pixel detector, we also successfully demonstrated a 64 × 64 image array with ~300 μm pixel pitch.

## Results and discussion

At the dark condition, when the DCS electrode is disabled and a working voltage is applied to the drain, the electrons are emitted from the source, then transport through the conduction channel, and eventually collected by the drain (Fig. 2a), the measured drain current is positive. This is the typical working mechanism of a photoconductive detector with two lateral terminals. In our design, we add an extra DCS electrode on the top. By applying a small voltage to it, the electrons emitted from the source will be attracted by the DCS electrode. Because the mobile electrons under dark are shunted, there are fewer electrons being collected by the drain, thus the dark current in the actual conduction channel is suppressed (Fig. 2b). In principle, there is a critical voltage (CV), under which the mobile electrons under dark can be completely shunted by the top DCS electrode. In this circumstance, the drain will not receive any mobile electrons and the dark current can be zero (Fig. 2c). If the applied voltage on the DCS electrode increases further, the mobile electrons will be extracted from the drain to the DCS electrode (Fig. 2d), the dark current collected by the drain will become negative.

In order to minimize the absolute value of dark current and obtain a high SNR, the DCS electrode can be biased with a CV, making dark current fully shunted. Under X-ray, the excess carriers are generated in the photo-active material, those X-ray induced electrons are drifted when interfaced with the electron transport layer (ETL)[16], and sensitize the lateral conduction channel, which means the photosensitive material generates an abundance of charges, these charges being captured by a transport layer with higher mobility can transport rapidly in the conduction and reinject and recirculate swift between metal contacts. Then, a photoconductive gain can be obtained. As a result, a strong X-ray photocurrent can be measured at the drain (Fig. 2e). It should be noted that the electric field applied by the DCS electrode will possibly attract some of the X-ray induced

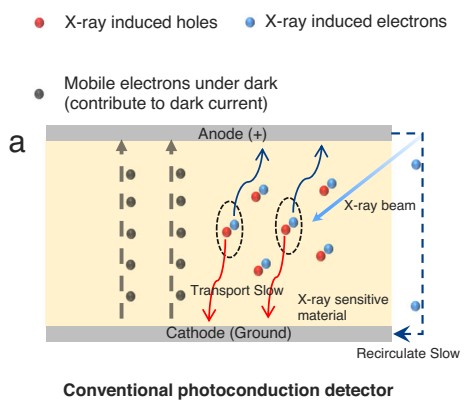
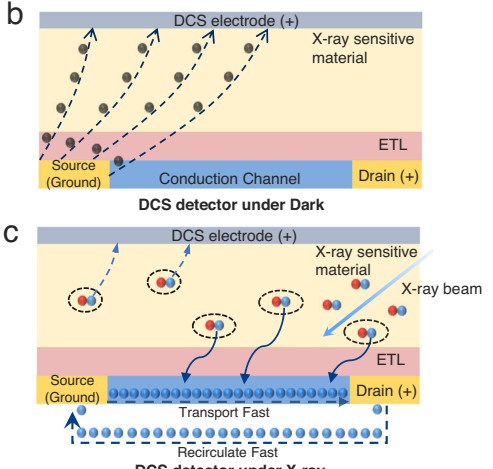

**Fig. 1 | Working mechanism of dark-current-shunting (DCS) detector and conventional photoconduction detector. a** Working mechanism of conventional photoconduction detector. The dark current and photocurrent are conducted in the same path and collected by the same electrodes. **b** Working mechanism of dark-current-shunting (DCS) detector. The electrons in the dark are emitted from the source and collected by the DCS electrode. **c** The X-ray induced electrons are generated from X-ray sensitive material and part of them

drifted into a conduction channel with high carrier mobility under a built-in electric field between X-ray sensitive material and electron transport layer (ETL), then collected by drain electrode under an electric field applied in the lateral conduction channel. Even some of the X-ray induced electrons will be attracted to DCS electrode, with the high photoconductive gain effect of the conduction channel, the photocurrent is still strong.

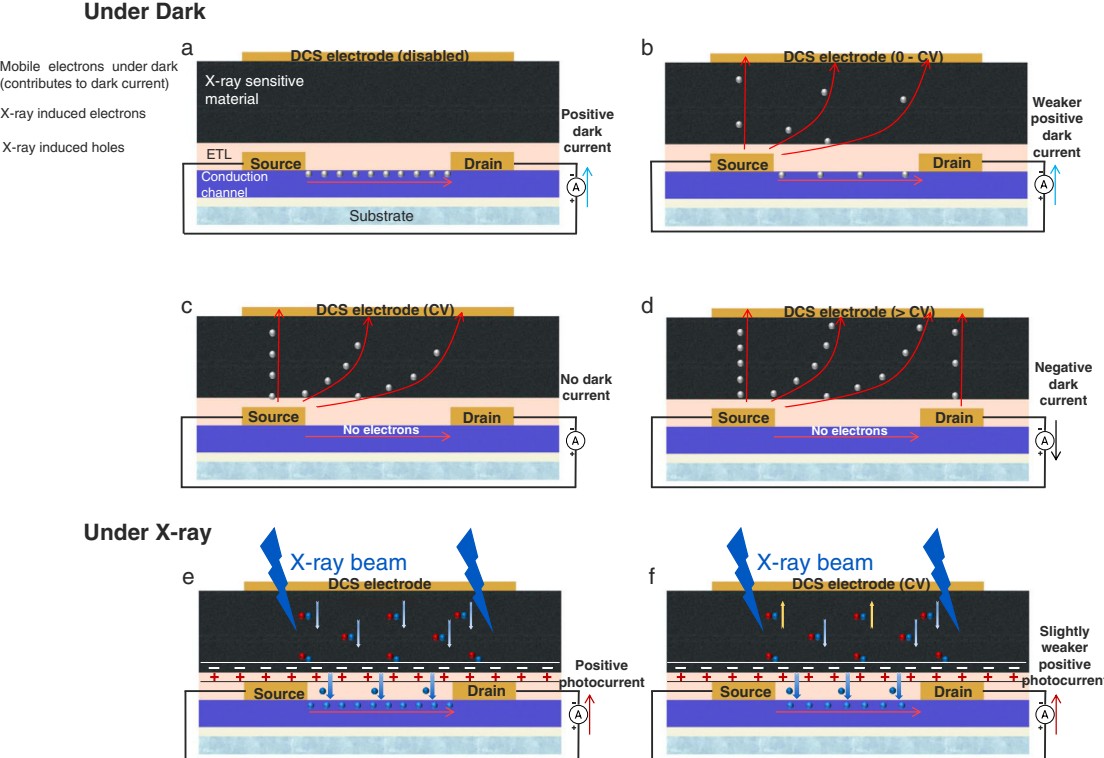

**Fig. 2 | Detailed working principles of dark-current-shunting (DCS) X-ray detector. a** Working principle at the dark condition and with DCS electrode disabled. The electrons emitted from the source transport through the conduction channel, and are collected by the drain, the DCS detector is working in a two-terminal mode similar as a photoconductive detector. **b** Working principle at the dark condition and with DCS electrode applied with a small bias. Some electrons emitted from the source are attracted by the DCS electrode, and the drain collects fewer electrons than the case described in Fig. 2a, leading to a smaller dark current. **c** Working principle at the dark condition and with DCS electrode applied with a CV. The electrons emitted from the source are all attracted by the DCS electrode, and the drain will not receive any electrons, leading to zero dark current in principle. **d** Working principle at the dark condition and with DCS

electrode applied with a bias larger than CV. Some electrons are extracted from the drain and collected by the DCS electrode, leading to a negative dark current. **e** Working principle at X-ray illumination and with DCS electrode disabled. The DCS detector is working in a two-terminal mode similarly as the photoconduction detector. **f** Working principle at X-ray illumination and with DCS electrode applied with a CV. In order to minimize the dark current, the DCS electrode should be biased with a CV. Under X-ray excitation, the photo-generated excess carriers are transported through the transport layer (ETL) and sensitize the conduction channel, eventually producing photocurrent at the drain (Source electrode is always contacted to the ground, drain electrode is applied with a working voltage. Working voltage and CV here are positive).

electrons and produce a lightly weaker photocurrent than the case with the DCS electrode disabled (Fig. 2f), but the overall SNR can still be significantly improved given the nearly completely suppressed dark current. The experimental evidence will be presented in the following section.

Following this design principle, we successfully fabricated DCS X-ray detectors with a structure as shown in Fig. 3a. Black-phase $FA_{0.92}Cs_{0.04}MA_{0.04}PbI_3$ perovskite was selected as an exemplary X-ray sensitive material in this study. Its X-ray diffraction (XRD) and photoluminescence (PL) are shown in Figs. S1 and 2. It has a density of ~4 g/cm³, providing a large linear attenuation coefficient of 10 cm⁻¹ to 40 keV X-ray (Figure S3). $In_2O_3$ is a kind of semiconductor material which has been widely used as the channel material for thin-film transistors. And it merits of low-temperature solution-processability. The trapped carriers on its interface can alter the conductivity of the $In_2O_3$ channel through capacitive coupling and high photoconductive gain can be obtained[39]. The primary functional layers of this device, perovskite and $In_2O_3$ are both processed by solutions. The band diagram of the device in the equilibrium at dark and under X-ray[39] are shown in Figure S4. The source electrode was always grounded, and the working voltage was applied to the drain during operation. As expected, first, the drain current decreased with the increased voltage applied to the DCS electrode (Fig. 3b).

At a CV of ~0.56 V, the drain current was almost fully suppressed to be nearly-zero. With further elevated voltage at the DCS electrode, the drain current turned to be negative and its absolute value continuously increased. As shown in Fig. 3b, with varied drain-source voltages (0.25 V, 0.5 V, and 0.75 V), we can always find a critical DCS voltage (0.30 V, 0.56 V, and 0.85 V, respectively) to shunt the dark current completely. Those detectors are operating with the matched DCS and drain-source voltages under X-rays, and the photocurrent is always positive under this working condition as shown in Fig. 3c. Given a fixed working voltage (drain-source voltage) of 0.5 V, we tested the X-ray photocurrents with 0 V, 0.56 V, and 1 V bias voltages applied to the DCS electrode (Fig. 3d). It can be seen that when a CV was applied to the DCS electrode, the drain current was decreased to nearly-zero, and the X-ray photocurrent was only slightly diminished, compared with the case of the disabled DCS electrode. The sensitivity of our DCS X-ray detectors reaches $1.3 \times 10^4$ μC $Gy_{air}^{-1}$ cm⁻² under the circumstance that the dark current is decreased to be nearly-zero, which is very much different from the typical 2-terminal X-ray photoconduction detectors (Fig. 3e). As the X-ray dose rate is increased, the photocurrent demonstrates a sublinear dependence on it. This reduction in sensitivity can be explained in terms of trap states present either in $In_2O_3$ or at the interface between the $In_2O_3$ and the underlying $SiO_2$ layer. Under high

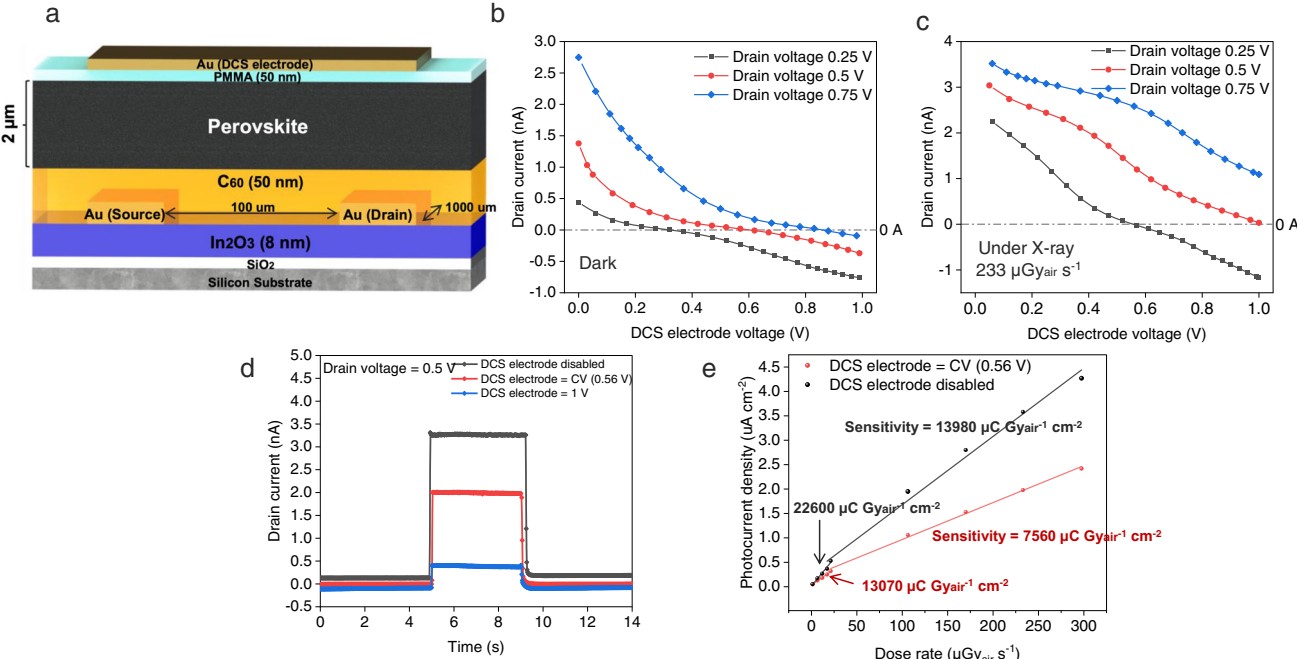

**Fig. 3 | Schematic, current–voltage characteristics and sensitivity of DCS X-ray detectors. a** Schematic of DCS X-ray detectors. **b** Current–voltage curves in terms of DCS electrode voltages and drain currents in dark conditions. The drain current decreases with increased DCS biases. **c** Current–voltage curves in terms of DCS electrode voltages and drain currents when exposed to X-ray. **d** X-ray generated pulse signals were illustrated when the DCS electrode was disabled, applied with a CV (0.56 V) and applied with a voltage larger than CV (1 V). The dose rate of X-ray was 233 $\mu Gy_{air}$ $s^{-1}$. **e** The sensitivity of the DCS detectors was calculated when the DCS electrode was disabled and applied with a CV (0.56 V).

illumination intensities the density of available trap states is reduced, resulting in saturation of the photoresponse[41]. Although our device structure appears to be as similar as a top-gated phototransistor, its functioning mechanism is different since there is no gate regulation effect with this very thin PMMA spacing layer (Figure S5). Some of the devices may use the thick PMMA layer as a dielectric layer, but their PMMA layer is too thick (~250 nm) to allow the current pass through[40]. Our PMMA layer is only ~50 nm, it's not insulative but may introduce a capacitance between DCS electrode and perovskite. The capacitance seems will not affect the final signal, which is collected in Drain electrode on the other side of the perovskite. Comparing the MHP transistor's transfer curves[42] to our device (Figure S5), there is no on/off state in our device when adjusting the DCS electrode's voltage. The PMMA was mainly used as a buffer to avoid unexpected chemical corrosions between perovskite and metal. The device without PMMA also shows similar dark current shunting behaviors when a voltage is applied to the DCS electrode. Its dark current can also be suppressed to nearly-zero when the voltage of the DCS electrode reaches a critical value of ~0.4 V (Figure S6), given a working voltage of 0.5 V. In a conventional X-ray photoconductive detector, the dark current is basically determined by the effective resistance between two terminals. Although metal halide perovskites have shown many attractive characteristics as next-generation X-ray detectors, its resistivity is usually too small and is fluctuating because of the ion drift. Our strategy provides a unique solution to suppress the dark current from the perspective of device design, and the dark current in principle is no longer limited by the resistivity of detector materials.

The dynamic response, SNR, lowest detection limit, and stability are evaluated in this secession. As shown in Fig. 4a, the device with a CV (0.56 V) applied to the DCS electrode clearly exhibits a much more stable signal current than the two-terminal photoconductor. The average photocurrent, dark current, and noise current are revealed.

The net signal current ($I_{net}$) was derived by subtracting the average photocurrent ($\bar{I}_{photo}$) by the average dark current ($\bar{I}_{dark}$).

$$I_{net} = \bar{I}_{photo} - \bar{I}_{dark} \qquad (1)$$

The noise current ($I_{noise}$) of was obtained by calculating the standard deviation of the dark current,

$$I_{noise} = \sqrt{\frac{1}{N}\sum_{i}^{N}\left(I_i - \bar{I}_{dark}\right)^2} \qquad (2)$$

The signal-to-noise ratio was calculated as:

$$SNR = I_{net}/I_{noise} \qquad (3)$$

After applying CV to the DCS electrode, the dark current and noise current are as low as 51.1 fA and 152 fA (Figure S7), respectively, approaching the lowest measurable current of a typical semiconductor analyzer or source-meter. As a result, the SNR was improved by two orders of magnitude, showing an excellent signal output under the X-ray pulse train (Fig. 4a). When the DCS electrode is disabled, the dark current gradually shifts from 0.615 nA to 0.986 nA in the pulse-train measurement. While with the DCS electrode biased with the CV, the dark current reveals almost no shift (Fig. 4b). The noise current and the SNR of a single pulse photocurrent are 22.5 pA and 147.6, respectively, while the DCS electrode is disabled (Fig. 4c). But after the DCS electrode is biased with CV (0.56 V), its noise current and the SNR have tremendously improved to 0.152 pA and 12,500, respectively (Fig. 4d). The improved quality of signal current can also be verified under very low dose rate X-ray illumination. At the dose rate of 166 $nGy_{air}$ $s^{-1}$ and 83 $nGy_{air}$ $s^{-1}$, the respective SNRs of the DCS detector are 36.18 and 26.97, significantly larger than the SNRs of photoconductor detectors,

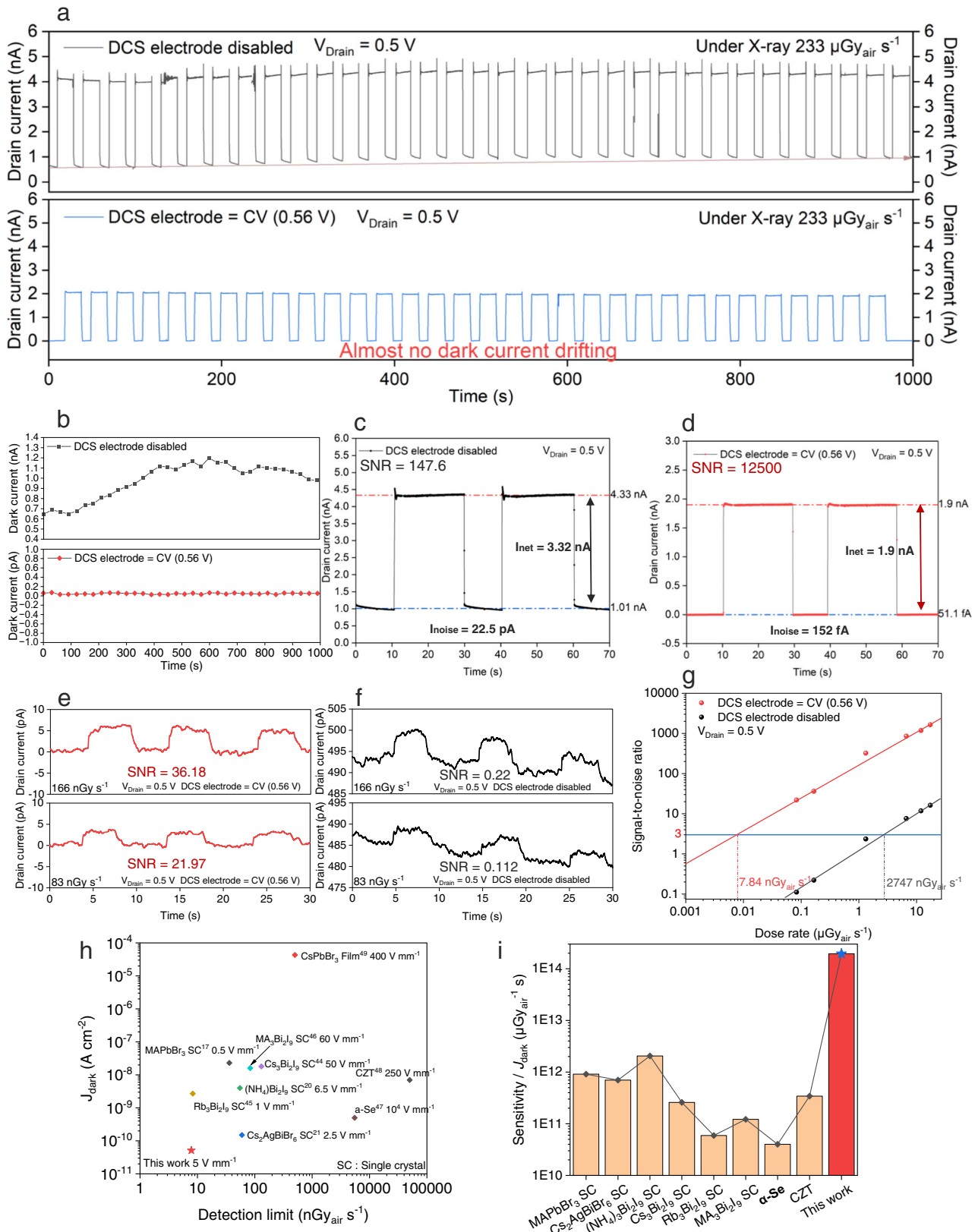

which are 0.22 and 0.112, respectively, as shown in Fig. 4e and f. The detection limit of the DCS detector (SNR = 3) is as low as 7.84 nGy$_{air}$ s$^{-1}$ (Fig. 4g), which is 705 times lower than the standard dosage for X-ray medical examination[43], and is 350 times lower than the control photoconductor detector. We compared our device with other reported state-of-the-art photoconductors in terms of dark current density,

detection limit[17,20,21,44–49] (Fig. 4h), as well as the sensitivity//$J_{dark}$ (Fig. 4i) which is a figure of merit to evaluate the SNR. It is shown that our DCS detector delivers very low dark current, low detection limit, and ultrahigh SNR, even outperforming many previously reported state-of-the-art single crystal-based photoconductors. The DCS method is a universal device strategy and can be applied in other perovskite

**Fig. 4 | Characterizations of pulse-train response, signal-to-noise ratio (SNR), and lowest detection limit. a** X-ray pulse-train response. By applying CV (0.56 V) to the DCS electrode, the signal-to-noise ratio (SNR), the pulse quality, and stability are obviously improved. There is almost no dark current drifting. **b** Characterization of dark current drifting. When the DCS electrode is disabled, the dark current gradually shifts from 0.615 nA to 0.986 nA. When the DCS electrode is biased with CV, the dark current reveals almost no shift during the pulse-train measurement. **c** The noise current and the SNR are 22.5 pA and 147.6, respectively, when the DCS electrode is disabled. **d** The noise current and the SNR are 0.152 pA and 12500, respectively, when DCS electrode is applied with a CV (0.56 V). **e** With the DCS electrode biased with a CV (0.56 V), the SNRs are 36.18 and 21.97 under respective X-ray dose rate of 166 nGy$_{air}$ s$^{-1}$ and 83 nGy$_{air}$ s$^{-1}$. **f** With the DCS electrode disabled, the SNRs are 0.22 and 0.112 under respective X-ray dose rate of 166 nGy$_{air}$ s$^{-1}$ and 83 nGy$_{air}$ s$^{-1}$. **g** Characterizations of detection limit. The detection limit (SNR = 3) of the DCS detector is as low as 7.84 nGy$_{air}$ s$^{-1}$, which is 350 times lower than the control photoconductor detector. The lowest detection limit was estimated by the reverse extension line to where the SNR = 3. **h** The comparison with respect to the dark current density and the detection limit between this work and other state-of-the-art photoconduction detectors is demonstrated. The applied electric fields are noted along with the detector materials. **i** The sensitivity/$J_{dark}$, a figure of merit to evaluate SNR, is illustrated to compared our DCS detector and other reported state-of-the-art photoconduction detectors.

composition and conduction channel materials. We fabricated MAPbBr$_3$, MAPbI$_3$, and FA$_{0.92}$Cs$_{0.04}$MA$_{0.04}$PbI$_3$ on In$_2$O$_3$ and IGZO conduction channels, the electron transport layer is C$_{60}$. The current–voltage curves in terms of DCS electrode voltages and drain currents of these devices were measured in the dark and under X-ray. It can be seen that the CV of these devices almost the same (0.35–0.4 V). All of the devices' dark current can be suppressed to 0 A by applying DCS voltage (Figure S8). When the DCS electrode is biased with CV, all of the devices demonstrate stable current baselines and high signal-to-noise ratio (Figure S9). Although depositing high-quality thick perovskite films is quite challenging, and is not the main focus of this study, we preliminarily demonstrate the thick film-based detectors, those devices are configured with 80 μm or 200 μm MAPbI$_3$/PMMA composite on top of SnO$_2$ electron transporting layer and In$_2$O$_3$ channel layer. The thick perovskite was fabricated by blade-coating with MAPbI$_3$ and PMMA polymer binding solution. When the DCS electrode is biased with CV, the dark current of thick film devices can also be suppressed to nearly zero and the device demonstrate stable current baselines and high signal-to-noise ratio. The CV seems not to be influenced by different compositions of materials, but will increase with larger conduction channel length, as well as thicker X-ray sensitive material (Figure S10), because stronger electric field are needed to attract the mobile electrons under dark to DCS electrode By using pure perovskite precursor solution, we varied the perovskite thicknesses between 300 nm to 2000 nm, we found that the thinner perovskite devices have lower sensitivity in lines with its different X-ray stopping powers (Figure S11), it implies that the built-in electric field between perovskite and C$_{60}$ might strong enough to extract X-ray induced electrons close to the surface of 2 μm perovskite, we suggest the DCS detector may perform better by increasing perovskite thickness further while maintaining high film quality. To preliminarily investigate its spatial resolvability in scanning-based X-ray imaging, we measured its modulation transfer function (MTF) by scanning the object with a stepping motor (Figure S12a). The used line pair mask plate is shown in Figure S12b, the minimum resolved line pair is 5.5 lp mm$^{-1}$ for a single-pixel device. We also compared our DCS detector with the typical vertically-stacked photoconductive detector with same perovskite composition and thickness (Figure S13a), whose current–voltage curves is shown in Figure S13b. Under the same 5 V mm$^{-1}$ electrical field, the dark current density is 72 nA cm$^{-2}$, that is more than a thousand time higher than our DCS device. Due to the high dark current and poor SNR, the control photoconductor device cannot generate neat signal output at the dose rate of 233 μGy$_{air}$ s$^{-1}$ (Figure S13c), and the dark current were shifting non-linearly from 1.05 nA to 1.23 nA during the pulse-train measurement, clearly, this sort of unpredictable baseline drift can't be simply removed by applying an offset voltage in the external circuit.

Lastly, we fabricated a proof-of-concept 64 × 64 matrix DCS detector array in an area of 2 cm × 2 cm and obtained the X-ray imaging. A schematic illustration of the DCS X-ray detector array is demonstrated in Fig. 5a, the sectional view is shown in Figure S14, the pixel pitch is ~300 μm. The solution-processed 64 × 64 matrix In$_2$O$_3$ TFTs back panel with a pixel density of 1024 per square centimeter is shown in Figure S17a. The microscopic images of the X-ray detector array and a single-pixel are given in Fig. 5b and c. The transfer curves and the output curves of a single In$_2$O$_3$ transistor in the array were measured as shown in Figure S15. We also calculated the linear electron mobility (Fig. 5d), the subthreshold swing (Figure S16a), the threshold voltage (Figure S16b) and the current on/off ratio (Figure S16c) of 400 randomly picked pixels and illustrated the distributions of those parameters. 50 nm thick of C$_{60}$ and 2 μm thick of perovskite were integrated with In$_2$O$_3$ TFTs. By using the Poly(perfluoroalkyl vinyl ether) (CYTOP) to cover the inactive region in the array, the perovskite precursor will not infiltrate into those areas and will only be deposited in the active region. This patterned-growth method contributes to a smooth and precisely covered perovskite layer (Figure S17b). Then a layer of Au (DCS electrode) was evaporated on top of the detector array (Figure S17c). The traditional a-Se array is fabricated by depositing a-Se on top of the a-Si TFT back panel, and then evaporating the top electrode as the common terminal of the photoconductor. Although our DCS single-pixel detector appears to have more complicated device structure, the fabrication of array detector is actually quite similar. We first fabricated the pixelized In$_2$O$_3$ TFT back panel, and then deposit the X-ray sensitive material (perovskite) onto the In$_2$O$_3$ TFT array and finally evaporate the top contact as the common DCS electrode. As for the readout circuit, the a-Se detector directly readout the charge (current) signal from drain electrode of the switching MOSFET, our device also readout the signal from drain electrode of the In$_2$O$_3$TFT. In addition, the analog data processing units including the preamplifier, Analog to Digital (AD) conversion can be shared with the conventional a-Si sensors. The preliminary stability test showed no observable degradations for a total received dose of 8.5 Gy$_{air}$, the measurement was conducted in ambient air (Fig. 5e). The X-ray imaging contrast is determined by the value of $I_{net}$/$I_{dark}$ (Fig. 5f). A 2 mm thick stainless steel plate of 'Qiushi Eagle' (Figure S17d) was placed on top of the array and the image was captured at the dose rate of 21.25 μGy$_{air}$ s$^{-1}$ (Fig. 5g, h). The drain was applied a bias of 0.5 V, the DCS electrode was biased with 0.56 V, and the bottom gate of the switching TFT was biased with 20 V. The contrast ratio is tremendously improved and a clear image is acquired once the DCS electrode is applied with the CV. In order to demonstrate its capability of detecting an invisible object, we made a coil spring in the capsule, a commonly used demonstration in X-ray imaging test (Fig. 5i)[3,18]. The coil spring (Fig. 5j) is not observable by naked eyes but can be well visualized by our DCS image array under 21.25 μGy$_{air}$ s$^{-1}$ X-ray irradiation (Fig. 5k).

In conclusion, a novel device structure with a shunting electrode to split the flows of dark and photo currents is proposed and demonstrated in this study. It allows a tremendous suppression of the dark current to be nearly-zero with only slightly weakening of the photocurrent. A record low dark current of 51.1 fA (51.1 pA cm$^{-2}$) and a noise current of 152 fA are experimentally demonstrated, the ratio of sensitivity/dark current representing the SNR of the device is outperforming previously reported photoconductive detectors by two orders of magnitude. In addition, our DCS detector shows no drift of baseline current that is often observed in typical perovskite photoconductive detectors. A DCS detector array with a pixel density of 1024

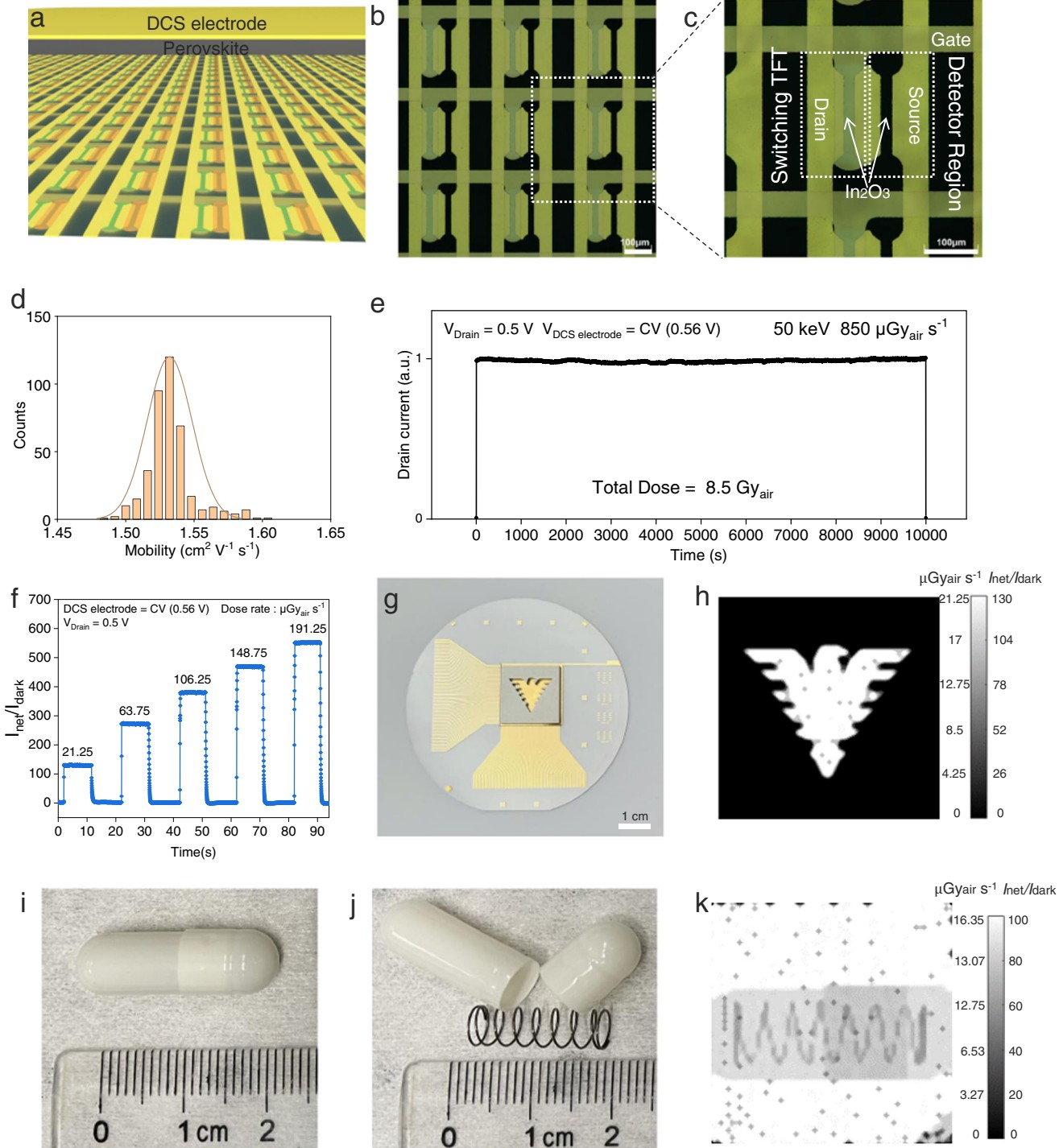

**Fig. 5 | DCS X-ray detector array and X-ray imaging. a** Schematic illustration of the DCS X-ray detector array. **b** Optical image of a fraction of detector array of nine pixels. **c**, The pixel pitch is 312.5 μm. The photosensitive area is 30 μm × 200 μm. **d** Electron mobility distribution of randomly selected 400 pixels. **e** Stability test of the DCS detector. The X-ray photon energy is 50 keV and the dose rate is 0.85 mGy$_{air}$ s$^{-1}$. The total received dose is 8.5 Gy$_{air}$. **f** The image contrast is determined the value of $I_{net}/I_{dark}$ at different X-ray dose rates. **g** The optical image of the 64 × 64 detector array and a 2 mm thick stainless steel mask plate with a hollowed-out figure of 'Qiushi Eagle'. **h** Obtained X-ray image while the DCS electrode is biased with CV (0.56 V). **i** Top view of the coil spring in the capsule. The coil spring cannot be seen by eyes. **j** Top view of the coil spring. **k** Obtained X-ray image of the coil spring in the capsule, with the DCS electrode biased with the CV (0.56 V).

per square centimeter is successfully demonstrated with both solution-processed X-ray active material (perovskite) and electronic channel material (In$_2$O$_3$). This device strategy is not only a new perspective to solve the challenging issue in metal-halide X-ray detectors, but also should be generally effective in other types of photodetectors and material systems that require low dark current and high SNR.

## Methods

### Materials

Indium nitrate hydrate (In(NO$_3$)$_3$·xH$_2$O, Aldrich, 99.999%) powder was dissolved in 2-methoxyethanol (2-ME) to prepare the In$_2$O$_3$ precursor solution (0.1 M). The IGZO precursor solution (0.1 M) was prepared by dissolving indium nitrate (In(NO$_3$)$_3$·xH$_2$O, 99.999%), gallium nitrate

$(Ga(NO_3)_3 \cdot xH_2O$, 99.999%), and zinc acetate ($Zn(CH_3COO)_2 \cdot 2H_2O$, 99.999%) in 2-methoxyethanol (2-ME). The molar ratio of the In:Ga:Zn in the IGZO precursor solution was 9:1:2. The ITO precursor solution (0.1 M) was prepared by dissolving indium nitrate and stannous chloride ($SnCl_2 \cdot 2H_2O$, 99.995%) in 2-methoxyethanol (2-ME). The molar ratio of the In:Sn in precursor solution was kept at 9:1. Then, acetylacetone and ammonium hydroxide were added as additives. Formamidinium iodide (FAI, >98.0%), Methylammonium iodide (MAI, >99.0%), Methylammonium bromine (MABr, >99.0%), Cesium iodide (CsI, >99.0%) and Lead(II) iodide ($PbI_2$, 99.99%), Lead(II) bromine ($PbBr_2$, 99.99%) were purchased from TCI. Fullerene-$C_{60}$ ($C_{60}$, 99.5%), Tin (IV) oxide ($SnO_2$, 99.99%) was purchased from Sigma-Aldrich Company Ltd. Temozolomide (DMSO, 99%), N,N-Dimethylformamide (DMF, 98%) and chlorobenzene (CB, 99.8%) were purchased from J&K Scientific. Polymethyl methacrylate (PMMA, 495 A2 2%) was purchased from Nippon Kayaku Co. Ltd. Poly (perfluoroalkenyl vinyl ether) (CYTOP) was purchased from Asahi glass company (Japan) and consists of CTL-809M (solute) and CT-Solv.180 (solvent).

## Single-pixel device fabrication

The X-ray detectors were fabricated with a bottom gate top contact (BGTC) structure. 100 nm $SiO_2$ was deposited on silicon ($P^{++}$) substrate by PECVD at 350 °C as a dielectric layer. After $O_2$ plasma treatment for 3 min, the $In_2O_3$, IGZO, ITO precursor solution filtered through a 0.2-µm syringe filter was spin-coated at 3000 rpm for 30 s on the substrate as a channel layer. After spin-coating, the $In_2O_3$, IGZO, ITO thin film was annealed at 100 and 200 °C for 1 min and 1 h, respectively. And the channel layer was patterned via a conventional photolithography process (AZ-1518 photoresist, 90 °C baking, AZ 300MIF developer) and etched in a mixed solution of hydrochloric acid and deionized water (1: 10, v-v) for 10 s. Next, Ni (3 nm) and Au (50 nm) were sequentially deposited as source/drain by electron-beam evaporation and patterned by lift-off photolithography (AR-P 5350, 105 °C baking, AR 300-26 developer), forming channel W/L of 1000/100 µm. Afterward, 50 nm of $C_{60}$ was thermally evaporated in a separate vacuum chamber ($<5 \times 10^{-4}$ Pa) in sequence. The $SnO_2$ solution was spin-coated by one step program at 4000 rpm for 30 s, and then annealed at 150 °C for 30 min. The $FA_{0.92}Cs_{0.04}MA_{0.04}PbI_3$ perovskite precursor (2.8 M) solution prepared by mixing CsI (29.1 mg) MAI (17.8 mg) FAI (443 mg) $PbI_2$ (1290.83 mg) in 600 µL DMF and 400 µL DMSO, the $FA_{0.92}Cs_{0.04}MA_{0.04}PbI_3$ perovskite precursor (2.0 M) solution prepared by mixing CsI (29.1 mg) MAI (17.8 mg) FAI (443 mg) $PbI_2$ (1290.83 mg) in 1120 µL DMF and 280 µL DMSO, the $FA_{0.92}Cs_{0.04}MA_{0.04}PbI_3$ perovskite precursor (1.4 M) solution prepared by mixing CsI (29.1 mg) MAI (17.8 mg) FAI (443 mg) $PbI_2$ (1290.83 mg) in 1600 µL DMF and 400 µL DMSO, the $MAPbI_3$ perovskite precursor (2.0 M) solution prepared by mixing MAI (317.94 mg) $PbI_2$ (922.02 mg) in 800 µL DMF and 200 µL DMSO, the $MAPbBr_3$ perovskite precursor (2.0 M) solution prepared by mixing MABr (223.94 mg) $PbBr_2$ (734.02 mg) in 800 µL DMF and 200 µL DMSO. The precursors were stirred at room temperature for 120 min and filtered with a 0.22 µm PTFE filter prior to use. 500 nm, 1 and 2 µm $FA_{0.92}Cs_{0.04}MA_{0.04}PbI_3$: 35 µL $FA_{0.92}Cs_{0.04}MA_{0.04}PbI_3$ precursor solution (2.8 M for 2 µm and 2.0 M for 1 µm, 1.4 M for 500 nm) was spin-coated on top of the $C_{60}$ layer by a two-consecutive step program at 1000 rpm for 10 s and 4000 rpm for 40 s (1.4 M for 30 s), the antisolvent CB was added at a 5-s countdown. The devices were immediately annealed on a hotplate at 100 °C for 10 min. 1 µm $MAPbI_3$: 35 µL $MAPbI_3$ precursor solution was spin-coated on top of the $C_{60}$ layer by one step program at 4000 rpm for 30 s, the antisolvent CB was added at a 15 s countdown. The devices were immediately annealed on a hotplate at 100 °C for 10 min. 1 µm $MAPbBr_3$: 35 µL $MAPbBr_3$ precursor solution was spin-coated on top of the $C_{60}$ layer by one step program at 4000 rpm for 30 s, the antisolvent CB was added at a 15 s countdown. The devices were immediately annealed on a hotplate at 100 °C for 10 min. 80 µm

and 200 µm $MAPbI_3$: MAI (477 mg) and PbI2 (1393 mg) dissolved in 3 mL GBL and stirred at room temperature for 120 min and filtered with a 0.22 µm PTFE filter. 300 µL solution was added to 3 mL Toluene in centrifuge tube, then centrifuged at 8000 rpm for 1 h. The liquid part was replaced by Toluene and then centrifuged at 8000 rpm for 1 h again. Afterward, the $MAPbI_3$ powder was taken out and dried out and then added into the PMMA and Toluene mixing solution and stirred at room temperature for 2 h. Then the $MAPbI_3$ precursor solution was blade coated on $SnO_2/In_2O_3$ TFT substrate. The distance of the blade from the substrate were 500 µm (film dried out was 80 µm) and 900 µm (film dried out was 200 µm) and then dried out in the air for a day. 50 nm of PMMA spin-coated by 4000 rpm for 40 s was coated above, then a 100 nm Au has evaporated above with a mask plate covered with the area of the drain, source and the conduction channel.

## Detector array fabrication

The glass substrate (3 inch) was cleaned by immersing into acetone, isopropanol, and deionized water for 10 min, respectively, with ultra-sonication. First, the back gate electrode region is patterned by using the standard lithography process. Next, the Ni/Au (5/30 nm) gate electrodes were evaporated by electron beam (Ei-5z, ULVAC), and then lifted off. Afterward, 100 nm $SiO_2$ gate insulator was deposited by plasma-enhanced chemical vapor deposition (PECVD; SAMCO Inc. PD-220NL) at 300 °C. And the prepared $In_2O_3$ precursor solution was spin-coated as the semiconductor layer at 3000 rpm for 30 s and pre-baked on a hotplate at 200 °C for 5 min. The photoresist (AZ-1518) etch mask layer was spin-coated on the $In_2O_3$ thin film (4000 rpm, 60 s) and patterned by a mask aligner (Karl Suss MA6 mask aligner). Subsequently, the $In_2O_3$ thin film at the unwanted area was removed by the mixed solution of hydrochloric acid and deionized water ($HCl:H_2O = 1:10$, v-v) for 10 s. And the photoresist was removed by rinsing with acetone, isopropyl alcohol, and deionized water in a sequential way. Then the patterned $In_2O_3$ thin film were annealed at 300 °C for 1 h. The Ni/Au drain/source electrodes (5/50 nm) were deposited and patterned, using the same process used for bottom gate electrodes, forming channel W/L of 150/20 µm. Finally, the SU-8 2000.5 photoresist (500 nm) was spin-coated and patterned by photolithography to protect the transistor. For patterning the perovskite area, the CYTOP was spin-coated onto the device and patterned by $O_2$ plasma (PC-300, SAMCO Inc.). Subsequently, 50 nm $C_{60}$ as electron transport layer was deposited by thermal evaporation with the patterned shadow mask in a separate vacuum chamber ($<5 \times 10^{-4}$ Pa). Afterward, the perovskite precursor solution prepared by mixing CsI (29.1 mg) MAI (17.8 mg) FAI (443 mg) $PbI_2$ (1290.83 mg) in 600 µL DMF and 400 µL DMSO was stirred at room temperature for 120 min and filtered with a 0.22 µm PTFE filter prior to use. 60 µL precursor solution was spin-coated on top of the $C_{60}$ layer by a two-consecutive step program at 1000 rpm for 10 s and 4000 rpm for 40 s, the antisolvent CB was added at a 5-s countdown. The devices were immediately annealed on a hotplate at 100 °C for 10 min. 50 nm of PMMA spin-coated by 4000 rpm for 40 s was coated above, then Ni/Au (5/50 nm) has evaporated above with a mask plate covered with the area of the drain, source, and the conduction channel.

## X-ray photocurrent measurement

The X-ray source was used as a commercially available Amptek Mini-X tube with an Ag target and 4 W maximum power output. The total X-ray dose was modulated by changing the current of the X-ray tube (5−80 µA), as well as the distance between the device and the X-ray source. The radiation dose rate was calibrated by using a radical ion chamber dosimeter. In the stability test, the energy of the used X-rays is 50 keV. On the other measurement, the energy of the used X-rays is 40 keV. The used output spectrum at different voltages is shown in Figure S18. The transfer curves, output curves, the voltage and current of drain and DCS electrode of the devices were regulated and

measured by Keithley 2636B sourcemeter. For realizing data collection of the 64 × 64 active-matrix array, a data acquisition system was constructed by three PXI source measure unit (NI PXIe-4138) and a PXI matrix switch module (NI PXIe-2531) assembled in a PXI Chassis (NI PXIe-1073). Flexible cables were connected to the source, drain, and DCS electrode of the multiplexed detector using anisotropic conducting paste. The images detected by the detectors array were processed and median filtered by Matlab R2020a.

## Characterization of the materials

The $FA_{0.92}Cs_{0.04}MA_{0.04}PbI_3$ films were made on ITO substrates. The pre-patterned ITO substrates ($1.5 \times 1.5$ cm$^2$) were bought from PsiOTec Ltd and ultrasonically cleaned with soap, deionized water, acetone, and IPA in succession for 10 min. The as-cleaned ITO substrates were treated with UV-O$_3$ for 30 min and transferred to a N$_2$-filled glovebox. The XRD measurements of $FA_{0.92}Cs_{0.04}MA_{0.04}PbI_3$ were performed with Rigaku D/Max-B X-ray diffractometer with Bragg–Brentano parafocusing geometry, a diffracted beam monochromator, and a conventional cobalt target X-ray tube set to 40 kV and 30 mA. The photoluminescence (PL) was detected by Ocean Optics QE Pro. The 540 nm excitation light was made by Pico Quant PDL 800-D.

## Data availability

All data generated or analyzed during this study are included in this published article (and its supplementary information file). Source data are provided with this paper.

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

## Acknowledgements

The authors acknowledge funds received from the Natural Science Foundation of China (62074136, 61874096, 62174138). This work is also supported by the Westlake Multidisciplinary Research Initiative Center (MRIC) Seed Fund (MRIC20200101), the Leading Innovative and Entrepreneur Team Introduction Program of Zhejiang (2020R01005, 2019R02007), Fundamental Research Funds for the Central Universities (2022LHJH01-04) and State Key Laboratory of Advanced Technology for Materials Synthesis and Processing at Wuhan University of Technology (2022-KF-3). We also thank Westlake Center for Micro/Nano Fabrication, the Instrumentation and Service Center for Physical Sciences (ISCPS), and the Instrumentation and Service Center for Molecular Sciences (ISCMS) at Westlake University for the facility support and technical assistance. We thank Dr. Zhong Chen at the Instrumentation and Service Center for Molecular Sciences (ISCMS) for his contribution to the measurement of In$_2$O$_3$ film.

## Author contributions

Y.(M.)Y. and B.Z. conceived the idea and supervised the experiment. P.J. and Y.T. designed and fabricated the device, and collected the data. P.J. analyzed the data and made the figures with the assistance of Y.T., D.L., and Y.W. Y.T., D.L., and Y.W. fabricated the 64 × 64 In$_2$O$_3$ TFT array back panel together and P.J. complete the following part of the array. P.J. wrote the first draft and Y.(M.)Y. revised the manuscript. P.R. fabricated the thick perovskite films. P.R., Y.Y., W.Z., C.Z., T.L., and K.L. helped the materials characterization. C.K. and X.L. provided insightful dicussions on the project and helped refine the manuscript.

## Competing interests

The authors declare no competing interests.
