## [Peer Review File · Nature Communications]

Realizing Nearly-Zero Dark Current and Ultrahigh Signal-to-Noise Ratio Perovskite X-ray Detector and Image Array by Dark-Current-Shunting StrategyREVIEWER COMMENTS

Reviewer #1 (Remarks to the Author):

Peng Jin, Yingjie Tang et al. report about a dark-current shunting method for pixelated X-ray imagers based on Metal halide perovskites as absorbing layer.

The manuscript is well prepared and of interest for researchers working in the specific field of X-Ray detection.

The title is appropriate and reflects in a good way the work described in the article. Also, the abstract reflects the content of the article.

The introduction clearly describes the state of the art and application fields of the presented topics and problem being investigated. The reduction of the dark current is indeed one of the major challenges for perovskite-based X-Ray detectors.

Figures are accurately placed and easy to understand and the references are placed accurately.

Results are novel and quite remarkable, however the perovskite layer thickness of 2 μm precludes its use in medical X-ray imagers. The limited X-ray absorption (few percent) for 2 μm thick perovskites is too low to be used in industrial applications. This makes the impact of the manuscript rather limited. For the pixel architecture described in this work, and its working principle, it is not obvious to have the same performance of dark current suppression by retaining at the same time high photocurrent collection, for perovskite thicknesses of few hundreds of μm . If the authors can demonstrate that a similar behavior is obtainable for perovskite thicknesses of 100 μm and above, then I would recommend the publication in this journal otherwise a journal with lower impact factor should be better suited.

Reviewer #2 (Remarks to the Author):

The manuscript investigates a new device structure strategy for MHP x-ray detectors at the purpose to decrease the dark current in metal halide perovskite x-ray detectors. The strategy is based on the use of a planar geometry for the detection of photoelectrons induced by the x-ray irradiation and by a third electrode that works as a blanking unit for the noise. The manuscript is generally well written and the author show some nice results in their configuration. I have few comments that the authors should try to insert in their discussion to allow the reader to better understand the potential of their work.

1- The device structure that they propose is substantially more complex than the structure used for other direct conversion devices such the one based on a-Se. They should comment on the consequence of this more complicated structure for the read-out electronics.

2- The energy of the used x-rays should be stated in the experimental section.

3- In the experimental results the active layer is 2 micrometers, however, the thickness ideally should be higher. The authors should comment on the effect of the active layer thickness on their device performance and voltages that would need to be apply.

4- The authors mentioned that in their opinion this is not a transistor structure, but as they have a

substantial PMMA layer this is certainly working as capacitor. I think the authors should better explain their point by comparing with phototransistors structures eventually reported in literature if there are any made with MHP.

Reviewer #3 (Remarks to the Author):

The authors report a perovskite based x-ray detector with a new device structure using dark shunting electrodes to suppress dark current and obtained a high SNR. The dark current values obtained are probably the lowest reported so far in literature. Another significant result is that the dark current is stable which is essential for practical devices. They also demonstrate x-ray imaging with detector array.

Although the results obtained are impressive, the working mechanism or the device physics part was not clear to me. For example, it was not clear why the “dark electrons” flow from the source through ETL and perovskite to the dark shunting electrode but the photogenerated electrons flow in the opposite direction. The authors mention that the photogenerated electrons in the perovskite drift at the ETL interface, so wouldn't there be a barrier for dark electrons flowing in the opposite direction? Perhaps an energy band diagram can be shown?

Also, I was not convinced if this device structure is generally applicable. Would it still work for another perovskite composition and another conducting channel?

Figures could be improved. Some figures have a lot of text in small font.

Other comments

Pg 3 “...the drain only receives X-ray-generated electrons but rejects dark electrons....” This sentence seems misleading. Could the authors clarify how the drain rejects dark electrons?

Pg 3 Fig 1. The photo-induced electrons and holes are enclosed in a dashed oval. Could the authors explain what it means?

Pg 5 “...those photo electrons are drifted when interfaced with the electron transport layer (ETL) and sensitizes the lateral conduction channel” It is not clear what the authors mean by sensitizes the lateral channel.

Perhaps the authors could use an energy band diagram in SI?

Also I would expect the effective field at the interface to be not strong and solely relying on the ETL interface to extract electrons seems like an inefficient strategy.

Pg 6 “The primary functional layers of this device, perovskite and In_2O_3 ” Could the authors explain what

kind of material In_2O_3 is why they chose it? This is also the first time they mention In_2O_3 . Perhaps they could mention somewhere in the beginning that they used In_2O_3 as the conductive channel because I was under the impression that it was all perovskite.

Pg 8. Fig 3e. Could the authors comment on why the sensitivity for low dose is higher than for high dose?

Pg. 10 “The detection limit of the DCS detector (SNR=3) is as low as $7.84 \text{ nGyair s}^{-1}$ ” Could the authors show a pulse for $7.84 \text{ nGyair s}^{-1}$?

Pg 12. “of dark current drafting” Typo

Pg. 14 Fig 5d Could the authors comment on the mobility values? $1.55 \text{ cm}^2/\text{V}/\text{s}$ seems low.

Could the authors mention what the variation in the critical voltage for different devices were?

Firstly, we very much appreciated the constructive comments and valuable suggestions from reviewers 1, 2 and 3. Now all the questions have been thoroughly considered and carefully answered. And hopefully, our responses can release your concerns.

The author's answers to Reviewer #1's comment:

Dear reviewer:

Thank you very much for your supportive comments and constructive suggestions. We have carefully read your reviews and noticed your concerns. Your question was answered as below.

The manuscript has been revised with fully consideration of your comments.

Reviewer's comments: Peng Jin, Yingjie Tang et al. report about a dark-current shunting method for pixelated X-ray imagers based on Metal halide perovskites as absorbing layer.

The manuscript is well prepared and of interest for researchers working in the specific field of X-Ray detection.

The title is appropriate and reflects in a good way the work described in the article. Also, the abstract reflects the content of the article.

The introduction clearly describes the state of the art and application fields of the presented topics and problem being investigated. The reduction of the dark current is indeed one of the major challenges for perovskite-based X-Ray detectors.

Figures are accurately placed and easy to understand and the references are placed accurately.

A0: Thanks so much for your comments that you feel this work is novel and important.

Q1: Results are novel and quite remarkable, however the perovskite layer thickness of 2 μm precludes its use in medical X-ray imagers. The limited X-ray absorption (few percent) for 2 μm thick perovskites is too low to be used in industrial applications. This makes the impact of the manuscript rather limited. For the pixel architecture described in this work, and its working principle, it is not obvious to have the same performance of dark current suppression by retaining at the same time high photocurrent collection, for perovskite thicknesses of few hundreds of μm . If the authors can demonstrate that a similar behavior is obtainable for perovskite thicknesses of 100 μm and above, then I would recommend the publication in this

journal otherwise a journal with lower impact factor should be better suited.

A1: We appreciate it very much that you feel those results are novel and remarkable. We certainly understand your concerns. The main purpose of our work is to demonstrate a common device strategy method to suppress dark current and obtain a high signal-to-noise ratio, there is certainly quite a lot of rooms to further improve the device performance by material engineering, and we are working on it certainly. Actually in my points of view, depositing high-quality thick film perovskite is still quite challenging, since there is a solubility limit for the perovskite precursor solution.

Our device strategy is very universal and can be applied in various material systems with different thicknesses, and compositions, we have demonstrated this in our revised manuscript (page 12, Figure S8). Thick X-ray sensitive materials can obtain stronger absorption of incident X-ray. But the synthesis of high-quality thick perovskite films is still a big challenge around the world. We have used the nearly saturated perovskite precursor solution in the first version of the manuscript, the 2 μ m perovskite film is the thickest we can fabricate by the typical precursor-based deposition. In this round, we referred to some previous works on thick perovskite films and added experiments to verify our DCS method in thick films, which was fabricated by the polymer binding method. [Adv. Eng. Mater., 18: 1189-1199] [IEEE Sensors Journal, vol. 7, no. 6, pp. 925-930, June 2007] [Adv. Funct. Mater. 2022, 32, 2110729] The 200 μ m and 80 μ m MAPbI₃/SnO₂/In₂O₃ TFT devices are shown below. The devices are fabricated by blade-coating the MAPbI₃ and PMMA binder (dissolved in PMMA/Toluene solvent, MAPbI₃:PMMA = 1:2) on SnO₂/In₂O₃ TFT. Because the Toluene solvent will largely dissolve the C₆₀ layer, we replace it with SnO₂, which is also an electron transport layer (ETL) for perovskite.

The perovskite film consists of MAPbI₃ and PMMA polymer binder, PMMA is an insulator, thus, the conductivity and sensitivity of such devices are relatively low. We measured the I-V curves of the 200 μm MAPbI₃ and PMMA polymer binder film-based photoconductor (Au/MAPbI₃ and PMMA polymer mixture film/Au) in the dark and under X-ray, both dark current and photocurrent are very low. At the voltage of 0.5V, the photocurrent is 25 pA (Figure R1).

Figure R1: Current-voltage curves of 200 μm MAPbI₃ photoconductor in the dark and under X-ray.

We measured the current-voltage curves in terms of DCS electrode voltages and drain

currents of these thick devices in the dark and under X-ray. For all the 80 μm and 200 μm devices we fabricated, all of the devices' dark currents can be suppressed to 0 A. We found that the Critical Voltage increased with the rise of the thickness of X-ray sensitive materials (Figure R2a below). The dark current of 200 μm device can also be suppressed to nearly zero with DCS method, the dark current baseline is stable and signal-to-noise ratio can also be very high (Figure R2c,d below). We have added those new results in our revised manuscript (page 12).

Overall, we believe our device strategy proposed in this study is very universal and can be applied in various material systems with different thicknesses.

Figure R2: Performance of thick perovskite film devices. a, Current-voltage curves in terms of DCS electrode voltages and drain currents of 80 μm and 200 μm MAPbI₃/SnO₂/In₂O₃

TFT in the dark. **b**, Current-voltage curves in terms of DCS electrode voltages and drain currents of 80 μm and 200 μm MAPbI₃/SnO₂/In₂O₃ TFT under 21.25 $\mu\text{Gy}_{\text{air}} \text{s}^{-1}$. **c**, Pulse-train response of 200 μm MAPbI₃/SnO₂/In₂O₃ TFT device under 21.25 $\mu\text{Gy}_{\text{air}} \text{s}^{-1}$. The DCS electrode is biased with CV. **d**, Pulse-train response of 200 μm MAPbI₃/SnO₂/In₂O₃ TFT device under 21.25 $\mu\text{Gy}_{\text{air}} \text{s}^{-1}$. The DCS electrode is disabled

The author's answers to Reviewer #2's comment:

Dear reviewer:

We really appreciate your reviews and valuable pieces of advice for our paper. We have carefully read your comments and noticed your concerns. All of your questions were answered below.

The manuscript has been revised with full consideration of your concerns and suggestions.

Reviewer's comments: The manuscript investigates a new device structure strategy for MHP x-ray detectors at the purpose to decrease the dark current in metal halide perovskite x-ray detectors. The strategy is based on the use of a planar geometry for the detection of photoelectrons induced by the x-ray irradiation and by a third electrode that works as a blanking unit for the noise. The manuscript is generally well written and the author show some nice results in their configuration. I have few comments that the authors should try to insert in their discussion to allow the reader to better understand the potential of their work.

A0: Thanks so much for your comments that you feel this work well written and show some nice results.

Q1: The device structure that they propose is substantially more complex than the structure used for other direct conversion devices such the one based on a-Se. They should comment on the consequence of this more complicated structure for the read-out electronics.

A1: Thank you for your careful review and recognition. And thanks for this great question! We fully understand your concerns and we'd like to illustrate this more clearly in the revised manuscript, and we have added related comments in our revised manuscript (page 16). Our device structure is shown in Supplementary Information Figure S13, and is shown below

(Figure R1a). It is true that the single photoconductive detector is very simple, but when they are integrated with the back panel, the TFT device is also needed. The general structure of the pixelized a-Se detection unit is shown in Figure R1b below [Proc. SPIE 5368, Medical Imaging 2004: Physics of Medical Imaging, (6 May 2004)]. As you can see, the traditional a-Se array is fabricated by coupling the a-Se material in the surface of back panel, which is prepared ahead typically with a-Si, and then evaporating the top common contact electrode. In our structure, the In_2O_3 TFT array-based back panel is also prepared ahead, the latter process is actually quite similar. Given that we have the In_2O_3 TFT back panel in hand, what we do next is to deposit the X-ray sensitive material (perovskite) onto the In_2O_3 TFT array back panel and then evaporate the top common DCS electrode. The difference in the fabrication process is mainly the manufacture of the back panel. If the In_2O_3 (or IGZO) TFT array is fully developed and well-manufactured by the company, there won't be too much difficulty in making our DCS method-based device and arrays. As we can know the IGZO-based backplane is starting to take over the a-Si TFT. As for the signal readout, the a-Se detector directly readout the charge (current) signal from Drain electrode of the switching MOSFET (as shown below, Figure R1b), our device also readout the signal from Drain electrode of the switching TFT (Figure R1a), the analog data processing units including the preamplifier, Analog to Digital (AD) conversion can be shared with the conventional a-Si sensors, actually, we are using the same ROIC chips purchased from ADI in our study for the X-ray image demonstration.

Another difference might be related to the pixel size, with the industrial facilities, the typical pixel size of a photoconductive a-Se detector is $\sim 100 \mu\text{m}$, in this study, we have demonstrated the X-ray imager with a pixel size of $\sim 300 \mu\text{m}$, with our in-house lab facilities, compare with the commercial X-ray detector's pixel size ($100 \mu\text{m}$), we think it is not very difficult for companies to use our DCS method to produce $100 \mu\text{m}$ pixel based X-ray detector array in the future.

One particular good thing of our device is that, for the typical X-ray detector array, we usually have to purposely design a big storage capacitance (C_{st}) in each pixel, which is parallel connected with the built-in capacitance (naturally formed by the parallel electrodes), to beat the dark current. Otherwise, the capacitance is too small that they can be easily

filled up by the dark current. In our DCS detector, the dark current can be suppressed to extremely low, thus, in principle, we don't need to put an extra parallel storage capacitance in each pixel, which would simplify the pixel circuit design.

Figure R1: a, Sectional view of a single-pixel in the DCS array. The DCS detector is in series with a switching TFT to control the on/off of a single-pixel. **b**, Sectional view of a single-pixel in the a-Se array.

Q2: The energy of the used x-rays should be stated in the experimental section.

A2: Thank you for your advice, we have added the energy of the used x-rays (tube voltage: 40 keV and 50 keV) in our experimental section. (page 23) The used output spectrum at different voltages is shown below, the figure was added in the SI. (Figure S17)

Q3: In the experimental results the active layer is 2micrometers, however, the thickness ideally should be higher. The authors should comment on the effect of the active layer thickness on their device performance and voltages that would need to be apply.

A3: We appreciate it very much for your advice. We certainly understand your concerns. The main purpose of our work is to demonstrate a common device strategy method to suppress dark current and obtain a high signal-to-noise ratio, there is certainly quite a lot of rooms to further improve the device performance by material engineering, and we are working on it certainly. Actually in my points of view, depositing high-quality thick film perovskite is still quite challenging, since there is a solubility limit for the perovskite precursor solution.

Our device strategy is very universal and can be applied in various material systems with different thicknesses, and compositions, we have demonstrated this in our revised manuscript (page 12, Figure S8). Thick X-ray sensitive materials can obtain stronger absorption of incident X-ray. But the synthesis of high-quality thick perovskite films is still a big challenge around the world. We have used the nearly saturated perovskite precursor solution in the first version of the manuscript, the 2 μm perovskite film is the thickest we can fabricate by the typical precursor-based deposition. In this round, we referred to some previous works on thick perovskite films and added experiments to verify our DCS method in thick films, which was fabricated by the polymer binding method. [Adv. Eng. Mater., 18: 1189-1199] [IEEE Sensors Journal, vol. 7, no. 6, pp. 925-930, June 2007] [Adv. Funct. Mater.

2022, 32, 2110729] The 200 μm and 80 μm $\text{MAPbI}_3/\text{SnO}_2/\text{In}_2\text{O}_3$ TFT devices are shown below. The devices are fabricated by blade-coating the MAPbI_3 and PMMA binder (dissolved in PMMA/Toluene solvent, $\text{MAPbI}_3:\text{PMMA} = 1:2$) on $\text{SnO}_2/\text{In}_2\text{O}_3$ TFT. Because the Toluene solvent will largely dissolve the C_{60} layer, we replace it with SnO_2 , which is also an electron transport layer (ETL) for perovskite.

The perovskite film consists of MAPbI_3 and PMMA polymer binder, PMMA is an insulator, thus, the conductivity and sensitivity of such devices are relatively low. We measured the I-V curves of the 200 μm MAPbI_3 and PMMA polymer binder film-based photoconductor (Au/ MAPbI_3 and PMMA polymer mixture film/Au) in the dark and under X-ray, both dark current and photocurrent are very low. At the voltage of 0.5V, the photocurrent is 25 pA (Figure R2).

Figure R2: Current-voltage curves of 200 μm MAPbI₃ photoconductor in the dark and under X-ray.

We measured the current-voltage curves in terms of DCS electrode voltages and drain currents of these thick devices in the dark and under X-ray. For all the 80 μm and 200 μm devices we fabricated, all of the devices' dark currents can be suppressed to 0 A. We found that the Critical Voltage increased with the rise of the thickness of X-ray sensitive materials (Figure R3a below). The dark current of 200 μm device can also be suppressed to nearly zero with DCS method, the dark current baseline is stable and signal-to-noise ratio can also be very high (Figure R3c,d below). We have added those new results in our revised manuscript (page 12).

Overall, we believe our device strategy proposed in this study is very universal and can be applied in various material systems with different thicknesses.

Figure R3: Performance of thick perovskite film devices. **a**, Current-voltage curves in terms of DCS electrode voltages and drain currents of 80 μm and 200 μm MAPbI₃/SnO₂/In₂O₃ TFT in the dark. **b**, Current-voltage curves in terms of DCS electrode voltages and drain currents of 80 μm and 200 μm MAPbI₃/SnO₂/In₂O₃ TFT under 21.25 μGy_{air} s⁻¹. **c**, Pulse-train response of 200 μm MAPbI₃/SnO₂/In₂O₃ TFT device under 21.25 μGy_{air} s⁻¹. The DCS electrode is biased with CV. **d**, Pulse-train response of 200 μm MAPbI₃/SnO₂/In₂O₃ TFT device under 21.25 μGy_{air} s⁻¹. The DCS electrode is disabled

Q4: The authors mentioned that in their opinion this is not a transistor structure, but as they have a substantial PMMA layer this is certainly working as capacitor. I think the authors

should better explain their point by comparing with phototransistors structures eventually reported in literature if there are any made with MHP.

A4: Thanks for this wonderful question, we totally understand your concerns, and we have added more discussions in our revised manuscript (page 9). We'd like to prompt that in our device, even without the PMMA layer, a similar device behavior is observed, and there is the dark-current-shunting effect as well (Supplementary Information Figure S6). We believe the very thin PMMA layer only acts as a protection layer and we don't find a capacitor effect in the photocurrent and dark current. Some of the devices may use the thick PMMA layer as a dielectric layer, but their PMMA layer is usually very thick. Our PMMA layer is only ~50 nm, it can hardly act as a dielectric layer. Besides, comparing the MHP transistor's transfer curves [Nat Electron 5, 78–83 (2022)] to our device (Supplementary Information Figure S5), there is no on/off state in our device when adjusting the DCS electrode's voltage.

The author's answers to Reviewer #3's comment:

Dear reviewer:

Thank you very much for your reviews and valuable suggestions for our paper. These suggestions really help us to refine our manuscript and make it more understandable for readers. We have carefully read your comments and noticed your concerns. All of your questions were answered below. And the manuscript has been revised with the entire consideration of your suggestions and concerns.

Reviewer's comments: The authors report a perovskite based x-ray detector with a new device structure using dark shunting electrodes to suppress dark current and obtained a high SNR. The dark current values obtained are probably the lowest reported so far in literature. Another significant result is that the dark current is stable which is essential for practical devices. They also demonstrate x-ray imaging with detector array.

A0: Thanks so much for your comments that you recognize the work's result about the low and stable dark current as well as the imaging with detector array.

Q1: Although the results obtained are impressive, the working mechanism or the device physics part was not clear to me. For example, it was not clear why the “dark electrons” flow from the source through ETL and perovskite to the dark shunting electrode but the photogenerated electrons flow in the opposite direction. The authors mention that the photogenerated electrons in the perovskite drift at the ETL interface, so wouldn't there be a barrier for dark electrons flowing in the opposite direction? Perhaps an energy band diagram can be shown?

A1: Thank you very much for your positive comments and very careful review. We understand your concerns and revised the working mechanism (Fig.1 in the manuscript). Your comments really promote us to think deeper and in more detail about the device's working principles, we have added a band diagram below, and put more discussion on the working mechanism, the energy levels come from our previous work measured with UPS [Small Methods 2022, 6, 2200500].

Firstly, with our device design, the X-ray photocurrent is actually decreased under DCS mode, compared with the control device without DCS voltages. But the actual conduction channel is the high-mobility In_2O_3 , not the perovskite, and there is a photoconductive gain of the In_2O_3 channel, therefore the decrease of X-ray photocurrent is not much, and the overall SNR is enhanced with several orders of magnitudes. This sort of gain is observed in many hybrid device structure. For the typical two-terminal photoconductor or photoresistor detector made of perovskite, the conduction channel is always perovskite and is usually without gain (unless there is trapped-induced photoconductive gain), we believe the photocurrent will be cut more if we put a similar DCS electrode.

Let us get back to our device structure, under the dark conditions, most of the dark electrons can be collected by the DCS electrode (positive), or at least they will no longer be received by the drain, which gives the zero-dark current. (In the manuscript page 4, Figure 1b; Page 10, Figure 3b; Page 14, Figure 4a). We believe this observation is solid.

We admit that this DCS bias could also affect the X-ray photocurrent since we indeed find a reduced photocurrent. (In the manuscript Page 14, Figure 4a) In order to better illustrate the detailed working mechanism under X-ray, we divided the device in two regions

separated by the red dash line. Those two regions have different external electrical fields, the source is grounded and the DCS and drain are both positive, (in our cases they are 0.56 and 0.5 respectively), the left region has much larger external electric field than the right region.

On the left of the red dash line: As shown in Figure b, between the Source electrode and DCS electrode, there is an externally applied voltage when the device is working, this external voltage will generate an electric field from DCS electrode to Source. The photogenerated electrons in this region move in the same direction with shunted dark electrons. Thus, the photocurrent decreases when the device is working, as you can see in Fig. 3e in the manuscript. But with the great photoconductive gain effect of the In_2O_3 conduction channel (high mobility), the sensitivity can still reach a high value.

On the right of the red dash line: The external electric field is much weaker in this region

(the DCS and Drains are both positively biased, and in our cases they are 0.56 and 0.5 respectively), and the built-in electric field is more favorable for pushing electron downward to the conduction channel. (Figure b). But with the built-in electric field between perovskite and C_{60} due to their energy band type-II alignment, you can see it from figure b, it will be more easier for the photogenerated electrons to be drifted from the perovskite towards the In_2O_3 . Those X-ray photo-charges mainly contribute to the observed drain photo-current with the help of the fast conduction channel.

Q2: Also, I was not convinced if this device structure is generally applicable. Would it still work for another perovskite composition and another conducting channel?

A2: We totally understand your concerns and fabricated the devices with different perovskite materials and other metal oxide conduction channels to demonstrate the generality of DCS method. The devices are shown below, we used $MAPbBr_3$, $MAPbI_3$ and $FA_{0.92}Cs_{0.04}MA_{0.04}PbI_3$ on In_2O_3 and IGZO conduction channel.

We measured the current-voltage curves in terms of DCS electrode voltages and drain currents of these devices in the dark and under X-ray (Figure R1). It can be seen that the Critical Voltage (CV) of these devices almost the same (0.35 – 0.4 V). All of the devices' dark current can be suppressed to 0 A by applying the DCS method.

Figure R1: Current-voltage curves in terms of DCS electrode voltages and drain currents of devices with 1 μm MAPbBr₃, MAPbI₃ and FA_{0.92}Cs_{0.04}MA_{0.04}PbI₃ on In₂O₃ and IGZO conduction channel in the dark and under X-ray. **a**, Current-voltage curves of 1 μm MAPbBr₃, MAPbI₃ and FA_{0.92}Cs_{0.04}MA_{0.04}PbI₃ on C₆₀/In₂O₃ TFT in the dark. **b**, Current-voltage curves of 1 μm MAPbBr₃, MAPbI₃ and FA_{0.92}Cs_{0.04}MA_{0.04}PbI₃ on C₆₀/IGZO TFT in the dark. **c**, Current-voltage curves of 1 μm MAPbBr₃, MAPbI₃ and FA_{0.92}Cs_{0.04}MA_{0.04}PbI₃ on C₆₀/In₂O₃ TFT under 21.25 $\mu\text{Gy}_{\text{air}} \text{s}^{-1}$. **d**, Current-voltage curves of 1 μm MAPbBr₃, MAPbI₃ and FA_{0.92}Cs_{0.04}MA_{0.04}PbI₃ on C₆₀/IGZO TFT under 21.25 $\mu\text{Gy}_{\text{air}} \text{s}^{-1}$.

We measured these devices' pulse train response when the DCS electrode is biased with CV (Figure R2). All of the devices demonstrate stable current baselines and high signal-to-noise ratio (pulse-train response of FA_{0.92}Cs_{0.04}MA_{0.04}PbI₃/C₆₀/In₂O₃ have already demonstrated in the manuscript). It can be seen that the DCS method is very general and can be applied in various devices with different material systems. We have added this comment in our revised manuscript (page 12).

Figure R2: Pulse-train response of different perovskite and conduction channel materials based devices. The DCS electrode are all biased with CV. **a**, Pulse-train response of MAPbBr₃/C₆₀/In₂O₃ TFT under 21.25 μGy_{air} s⁻¹. **b**, Pulse-train response of MAPbI₃/C₆₀/In₂O₃ TFT under 21.25 μGy_{air} s⁻¹. **c**, Pulse-train response of MAPbBr₃/C₆₀/IGZO TFT under 21.25 μGy_{air} s⁻¹. **d**, Pulse-train response of MAPbI₃/C₆₀/IGZO TFT under 21.25 μGy_{air} s⁻¹. **e**, Pulse-train response of FA_{0.92}Cs_{0.04}MA_{0.04}PbI₃/C₆₀/IGZO TFT under 21.25 μGy_{air} s⁻¹.

Instead of using the DCS method to solution-processed devices, we also tried to apply it in sputtered IGZO-based devices. We are not the experts on sputtering IGZO, but we did try to do it here with our own sputtering, honestly, the TFT device is not much better than our solution-processed devices. As shown below, the dark current of the MAPbBr₃/SnO₂/IGZO (Sputtered) TFT can also be suppressed to 0 A with the DCS method.

Q3: Figures could be improved. Some figures have a lot of text in small font.

A3: Thank you very much for your advice. The small font has been improved in the figures.

Other comments

Q4: Pg 3 "...the drain only receives X-ray-generated electrons but rejects dark electrons..."

This sentence seems misleading. Could the authors clarify how the drain rejects dark electrons?

A4: Thank you very much for your careful review and noting it. This is a wrong statement and we have corrected the "rejects" to "will not receive" in the manuscript. "Not receive" means that there will no dark electrons reach the drain electrode and be collected.

Q5: Pg 3 Fig 1. The photo-induced electrons and holes are enclosed in a dashed oval. Could the authors explain what it means?

A5: We totally understand your concerns. The photo-induced electrons and holes are enclosed in a dashed oval means an electron-hole pair generated together under the X-ray illumination, and the electrons and holes in a dashed oval are used as examples to illustrate the motion of whole electrons in the material.

Q6: Pg 5 "...those photo electrons are drifted when interfaced with the electron transport

layer (ETL) and sensitizes the lateral conduction channel” It is not clear what the authors mean by sensitizes the lateral channel.

Perhaps the authors could use an energy band diagram in SI?

A6: We totally understand your concerns and we’d like to make them clear in our revised manuscript (page 6). Sensitization is the process of generating charge carriers in the semiconductors who cannot absorb X-ray and generate hole-electron pairs themselves (Such as In_2O_3). These semiconductors may not be able to generate charges under X-ray but have superior mobility to transport and recirculate charges and greatly amplify the photocurrent. Thus, we use a sensitizer (Perovskite) to help generate charges under the light (X-ray) and extract them to these semiconductors (In_2O_3). The photosensitive material generates an abundance of charges, these charges then being captured by a transport layer with superior mobility (we call it conduction channel). Then, a much higher photoconductive gain can be obtained. This method allows us to use separate layers for charge photogeneration and transport to enhance photoconductive gain. Many researchers have used such sensitizing method to obtain great high photosensitivity of photodetectors. [ACS Appl. Mater. Interfaces 2019, 11, 36880–36885] [Adv. Mater. 2015, 27, 6885–6891] [Adv. Mater. 2021, 33, 2101717]. In addition, we also added a band diagram in SI.

Q7: Also I would expect the effective field at the interface to be not strong and solely relying on the ETL interface to extract electrons seems like an inefficient strategy.

A7: Indeed, we understand your concerns. In this kind of lateral device structure, the

electrons extraction seems inefficient, but the core advantage of such hybrid lateral structure is the conduction channel has a great photoconductive gain, the captured carriers by conduction channel can transport rapidly in the conduction channel and reinject and recirculate swift between metal contacts. The amount of charges passing through the cross section of a conductor per unit of time is much greater. Therefore, these carriers in the conduction channel can offer much stronger photocurrent than in the X-ray-sensitive materials. This gain effect can greatly amplify the photocurrent signal with superior mobility of the conduction channel. Even the electrons extraction is inefficient, the conduction channel can amplify the photocurrent signal thousands or ten thousands of times with the photocurrent gain effect [Adv. Funct. Mater. 2020, 30, 1903907]. Thus, the finally obtained photocurrent can be much higher than the device only with photosensitive materials. What's more, the addition of ETL like C_{60} can be an effective way to help extract electrons from perovskite, and this has been demonstrated in other research [ACS Appl. Mater. Interfaces 2018, 10, 50, 44144–44151]. Our finally obtained photocurrent sensitivity still reaches high, as you can see that our detector's sensitivity can obtain $2 \times 10^4 \mu\text{C Gy}_{\text{air}}^{-1} \text{cm}^{-2}$. Only when the DCS electrode is working does the sensitivity decrease a little.

Q8: Pg 6 "The primary functional layers of this device, perovskite and In_2O_3 " Could the authors explain what kind of material In_2O_3 is why they chose it? This is also the first time they mention In_2O_3 . Perhaps they could mention somewhere in the beginning that they used In_2O_3 as the conductive channel because I was under the impression that it was all perovskite.

A8: We appreciate your comments and we have added the illustration in our revised manuscript (page 7). In_2O_3 is a kind of n-type semiconductor material which has been widely used as the channel material for thin-film transistors [Appl. Phys. Rev. 3, 021303 (2016)]. And it merits of low-temperature solution-processability. In this work, we used In_2O_3 and perovskite as transport layer and sensitizer layer, respectively, to construct the hybrid photodetector. Owing to the superior mobility of In_2O_3 thin film, it can accelerate the transport speed of photo-induced carriers generated in the perovskite. In addition, the trapped carriers on the interface can alter the conductivity of the In_2O_3 channel through

capacitive coupling and higher photoconductive gain can be obtained which have been well demonstrated in our prior work [Small Methods 6, 2200500 (2022)] [Adv Mater 27, 6885-6891 (2015)]. And the hybrid structure can also be effective for perovskite/graphene and perovskite/IGZO and other different material combinations [Adv Mater 27, 41-46 (2015)] [Adv Mater 32, e1907527 (2020)].

Q9: Pg 8. Fig 3e. Could the authors comment on why the sensitivity for low dose is higher than for high dose?

A9: Yes, we'd like to illustrate it in our revised manuscript (page 8). It is actually quite a common phenomenon in many reported perovskite X-ray detectors. As the X-ray dose rate is increased, the photocurrent demonstrates a sublinear dependence on it. This reduction in sensitivity can be explained in terms of trap states present either in In_2O_3 or at the interface between the In_2O_3 and the underlying SiO_2 layer. The photocurrent mainly comes from the photo-induced electrons filling the trap states in metal oxide and changing the metal oxide's conductivity, under high illumination intensities the density of available trap states is reduced, resulting in saturation of the photoresponse. [Nature Nanotechnology 8, 497-501 (2013)] [Adv. Funct. Mater. 2019, 29, 1808182]

Q10: Pg. 10 "The detection limit of the DCS detector (SNR=3) is as low as $7.84 \text{ nGy}_{\text{air}} \text{ s}^{-1}$ "
Could the authors show a pulse for $7.84 \text{ nGy}_{\text{air}} \text{ s}^{-1}$?

A10: We understand your concerns, but we are very sorry that because the $7.84 \text{ nGy}_{\text{air}} \text{ s}^{-1}$ is extremely low, in our experiment condition and with the limitation of our equipment, we can not demonstrate a pulse for $7.84 \text{ nGy}_{\text{air}} \text{ s}^{-1}$. But you can see that our detector under $83 \text{ nGy}_{\text{air}} \text{ s}^{-1}$ (the lowest dose that we believe we can reliably measure with our ion chamber.) shows a much higher photocurrent than $3 \times I_{\text{noise}}$ (152 fA), and the SNR also much higher than 3. In addition, this method to calculate the detection limit has been widely used before. [Nat Commun 12, 5258 (2021)] [Adv. Mater. 2021, 33, 2101717]. We agree with reviewers that it is not very solid to use the reverse extension line to estimate the lowest detection limit, but we have tried our best and give the pulse response at $83 \text{ nGy}_{\text{air}} \text{ s}^{-1}$ showing SNR

of honestly,

Q11: Pg 12. “of dark current drafting” Typo

A11: Thank you for your careful review and mention the mistake. We have corrected it to “of dark current drifting”.

Q12: Pg. 14 Fig 5d Could the authors comment on the mobility values? 1.55 cm²/V/s seems low.

A12: We understand your concerns. The In₂O₃ possesses a high intrinsic electron concentration and high mobility features [Journal of Physics and Chemistry of Solids 38, 819-824 (1977)] [Adv Mater 27, 7168-7175 (2015)]. In this work, we used solution-processed method to fabricate the In₂O₃ thin film transistor as the switching backplane for addressing the photodetector sensors. And the linear mobility of the In₂O₃ transistor was calculated according to the transfer curve by the following equation:

$$\mu_{lin} = \frac{dI_D}{dV_{GS}} \times \frac{L}{W} \times \frac{1}{C_{ox} \times V_{DS}}$$

However, the high contact resistance ($R_{contact}$) between the Ni/Au source/drain (S/D) electrodes and In₂O₃ thin film will hinder the extracted mobility value based on the transfer curves [IEEE Transactions on Electron Devices 66, 5166-5169 (2019)]. To illustrate this point, we fabricated two In₂O₃ transistors with Ni/Au and Al source/drain electrodes, respectively, for comparison as shown in the below figure R3a and b. The extracted linear mobility of the In₂O₃ transistor with Al S/D electrodes was 8.23 cm²/V·s which is higher than that with Ni/Au S/D electrodes (1.53 cm²/V·s).

Although the higher $R_{contact}$ affect the extracted mobility value, it can be conducive to reduce transistor off current (I_{off}) which is important for reducing the signal crosstalk among sensing units. In addition, using gold as the electrode can keep more stable device performance, because aluminum is easy to react with perovskite and cause device damage.

Figure R3: **a**, Transfer curve of In_2O_3 transistor with Ni/Au and **b**, Al source/drain electrodes.

Q13: Could the authors mention what the variation in the critical voltage for different devices were?

A13: Thank you very much for your advice. Yes, we'd like to mention this and have added the comments in our revised manuscript (page 12). With the results of current-voltage curves in terms of DCS electrode voltages and drain currents of same thick devices with different X-ray sensitive layer (contribute to most of the thickness of the device) and different thick devices with same X-ray sensitive material (Figure R4). The critical voltage (CV) seems will not be influenced by different compositions of materials, but will increase with greater distance with conduction channel, which is also the thickness of X-ray sensitive material. Thus, we propose with the increase of the distance with conduction channel, the applied electric field of DCS electrode can be weaker around the conduction channel, and thus have weaker force to attract the dark electrons. It need stronger electric field to attract the dark electrons to DCS electrode and suppress the dark current to 0 A.

Figure R4: **a**, Current-voltage curves of $1\mu\text{m}$ MAPbBr_3 , MAPbI_3 and $\text{FA}_{0.92}\text{Cs}_{0.04}\text{MA}_{0.04}\text{PbI}_3$ on

C₆₀/In₂O₃ TFT in the dark. **b**, Current-voltage curves in terms of DCS electrode voltages and drain currents of 80 μm and 200 μm MAPbI₃(PMMA binder)/SnO₂/In₂O₃ TFT in the dark.

REVIEWER COMMENTS

Reviewer #1 (Remarks to the Author):

Peng Jin, Yingjie Tang et al. improved the quality of the manuscript substantially and fully addressed my main concern related to the thickness of the absorbing perovskite layer. In the revised manuscript the thickness has been increased two orders of magnitudes up to 200 μm which is already significant, considering e.g., X-ray absorption in mammography application. The authors demonstrated the validity of their concept of dark current reduction for thick absorbing layers (200 μm) in a similar way as reported earlier for thin layers (2 μm). The signal to noise ratio clearly suffers from increasing the absorbing layer thickness, however it should not preclude publication. A trade-off between X-ray absorption, dark current and signal height must be found accurately if industrial applications are envisioned. I suggest the publication of the revised manuscript as is.

Reviewer #2 (Remarks to the Author):

The authors have made a very professional and careful work in answering the questions posed by the reviewers and in amending the manuscript. I'm of the opinion that the current version is of high quality and should be therefore published in Nature Communication.

Reviewer #3 (Remarks to the Author):

I am satisfied with most of the authors' answers. I have three other concerns:

First, in the revised version the authors mention "great photoconductive gain" several times. Can they get a rough estimation of the gain factor? If they have mobility and lifetime of the channel, then they can calculate this.

The disadvantage of having photoconductive gain is lower speed. Can the authors estimate the pulse rise/fall time?

Second the authors mention that the perovskite photoconductive layer has "low mobility" and the In_2O_3 channel has "superior mobility". For In_2O_3 they measured a mobility of 1.55 $\text{cm}^2/\text{V}\cdot\text{s}$. In the literature they cite (Adv Mater 27, 7168-7175 (2015)) 8 $\text{cm}^2/\text{V}\cdot\text{s}$ is reported. But the typical mobility values for polycrystalline perovskite reported in lit (0.1-10 $\text{cm}^2/\text{V}\cdot\text{s}$) are in the same range.

My third concern is about the thickness. I understand that the novelty here is the device structure and they have not optimized the thickness, interlayers, etc however do they even need 2 microns? The authors mention that the xray generated carriers drift under a built-in electric field between X-ray

sensitive material and electron transport layer. In my opinion, the electric field at the ETL interface is not strong and I'm not sure if the depletion region extends 2 microns to the top DSC electrode. I would expect the devices to be limited by charge carrier extraction. There is not much benefit of absorbing more xrays by making thicker films if they can't extract the carriers generated. Moreover, I would expect thicker devices to be worse. Since xray absorption decreases exponentially, for thicker devices most of the carriers generated close to the top surface from where the xray is illuminated might not reach the interface where extraction is more efficient.

Other minor comments:

Line 128 and 133 CV has been defined earlier. The authors could avoid using acronyms for terms that have not been used many times such as WV, PCD. They are not commonly used acronyms and makes it difficult to read

Line 172 "Our PMMA layer is only ~50 nm, it cannot act as a dielectric layer."
Thinner layers have higher capacitance than thicker layer.

Line 216 "The detection limit of the DCS detector 217 (SNR = 3) is as low as 7.84 nGyair s⁻¹(Fig. 4g),"
The authors should mention how they obtained this number in the manuscript (used reverse extension line to estimate the lowest detection....?)

Line 242 "To preliminarily investigate its spatial resolvability in scanning-based X-ray imaging, we measured its modulation transfer function(MTF) by scanning the object (Figure S11a). The used line pair mask plate is shown in Figure S11b, the minimum resolved line pair is 5.5 lp mm⁻¹."
Could the authors show the scanned x-ray image of the object and include it in SI? Was this a single pixel device? MTF for the detector array would be more relevant. Did they measure it?

Line 255 "of dark current drifting" Typo

Figure still have a lot of text in small font. They could put only the most relevant/important text in the figure and the description could go in the caption or in the main text.

For eg, they could remove or reduce in fig 2. direction of dark electron motion, photoconductive gain. In fig 4 almost no shift of dark current drifting
Fig 3a text too small

Reviewer #1:

Comments: Peng Jin, Yingjie Tang et al. improved the quality of the manuscript substantially and fully addressed my main concern related to the thickness of the absorbing perovskite layer. In the revised manuscript the thickness has been increased two orders of magnitudes up to 200 μm which is already significant, considering e.g., X-ray absorption in mammography application. The authors demonstrated the validity of their concept of dark current reduction for thick absorbing layers (200 μm) in a similar way as reported earlier for thin layers (2 μm). The signal to noise ratio clearly suffers from increasing the absorbing layer thickness, however it should not preclude publication. A trade-off between X-ray absorption, dark current and signal height must be found accurately if industrial applications are envisioned. I suggest the publication of the revised manuscript as is.

Answer: Thank you very much for your recognition of our work. And thank you very much for your careful review and suggestions to make the manuscript better. We appreciate your great contributions.

Reviewer #2:

Comments: The authors have made a very professional and careful work in answering the questions posed by the reviewers and in amending the manuscript. I'm of the opinion that the current version is of high quality and should be therefore published in Nature Communication.

Answer: Thank you very much for your recognition of our revision. And thank you very much for your careful review and suggestions to improve our manuscript. We appreciate your great contributions.

Reviewer #3:

Comments: I am satisfied with most of the authors' answers. I have three other concerns:

A0: Thank you very much that you are satisfied with our answers. It's our pleasure to hear those additional constructive advice and suggestions from you. They indeed help us to improve our work!

Q1: First, in the revised version the authors mention "great photoconductive gain" several times. Can they get a rough estimation of the gain factor? If they have mobility and lifetime of the channel, then they can calculate this.

The disadvantage of having photoconductive gain is lower speed. Can the authors estimate the pulse rise/fall time?

A1: Very nice suggestions. Your advice with respect to the gain factor and the speed of this kind of device has really aroused us to think deeper about how the photoconductive gain generates and how it connects to the device's performance. The "great" might be a little exaggerated the Gain factor, and we revised it as "high".

As for the gain factor, the total gain produced by Heterojunction X-ray Phototransistors can be calculated as [Adv. Mater. 2021, 33, 2101717] [Adv. Funct. Mater. 2019, 29, 1900234]:

$$G = I_s E_{e-h} / \epsilon D m e$$

Where I_s is the X-ray signal current (4 nA), E_{e-h} is the EHP creation energy given by an empirical mode, [Nucl. Instrum. Methods Phys. Res., Sect. A 2006, 565, 637.] $E_{e-h} = 1.43 + 2E_g$ (E_g is the bandgap of perovskite, 1.55 eV), ϵ is the fraction of absorbed photons (0.1 at 40 keV X-ray), D is the dose rate ($233 \mu\text{G}_{\text{air}} \text{s}^{-1}$), m is the mass of perovskite (2.4×10^{-8} g), and e is the elementary charge. Note that the total gain is composed of impact ionization gain and photoconductive gain, in which the former is determined by the ratio of X-ray photon energy E_{ph} to E_{e-h} .

Under the 40 keV X-ray. The overall gain factor (including ionization and photoconductive) is calculated as $\approx 3.24 \times 10^4$.

If using the mobility and lifetime of the channel to calculate the photoconductive gain, the gain factor can be calculated as:

$$G_P = \frac{t_r}{t_t} = \frac{t_r}{L^2} \mu V_{DS}$$

where μ is mobility (1.55 cm²/V s) and V_{DS} is applied drain-source voltage (0.5 V), L is the channel length (20 μm) and t_r is the carrier lifetime of the channel (2.041 ms). The carrier lifetime of the channel was measured by time-resolution photoluminescence spectrum (TRPL):

Figure R1. TRPL of In₂O₃ film. Its average carrier lifetime is calculated as 2.041 ms.

As you can see, the metal oxide's lifetime is relatively long. The unique oxygen-sensitized photoconduction mechanism has allowed the photoconductive gain of metal oxide to reach an extremely high level, which is several orders of magnitude higher than the conventional thin film detectors. According to the widely adopted photoconduction model, the long carrier lifetime of metal oxide is proposed to be the origin of high-gain transport. [Nanoscale, 2013, 5, 6867-6873] However, the very long carrier lifetime of metal oxide has aroused some

unpleasant characteristics, such as slow response time, especially the persistence photocurrent (PPC) effect, which has been frequently founded in metal oxide-based phototransistors. [*Adv. Mater.* 2015, **27**, 6885–6891]

The Gain factor is calculated as ≈ 400 . As for the perovskite, its lifetime is estimated as 100 ns the Gain factor of 2 μm perovskite photoconductive detector (under the same applied electric field) with the same mobility is calculated as ≈ 0.2 , which is much smaller than our type of devices.

As for your next concern, we totally agree with your words that “The disadvantage of having photoconductive gain is lower speed.” The slower recombination of the carriers that took part in the transportation in the channel, the higher gain. Thus, the high gain and fast speed generally cannot be obtained simultaneously. We measured the rise and fall time of our device under the laser beam pulse. The DCS electrode is applied with CV. The rise and fall time are 23 ms and 31 ms, respectively:

Figure R2. Laser beam pulse response of the device. The rise and fall time are 23 ms and 31 ms, respectively.

Such a response speed can meet the general 30 Hz dynamic monitoring application. But as for X-ray detection, the commercially used X-ray detectors, such as a-Se detector, are fabricated in a very thick geometry, to obtain stronger absorption to X-ray. But such a thick film will definitely sacrifice the response speed to the X-ray. And in most practical cases, a-Se detector can only be used in static X-ray imaging. Even for the potential dynamic imaging applications, 30 Hz can be satisfied in most situations. In our opinion, at least at this moment, for the direct perovskite X-ray detector, the dynamic response is not the first priority, since the competing technology (a-Se) is not used for dynamic X-ray imaging as well. Additionally, for those reported perovskite X-ray detectors with typical vertical geometry, the response time is also not much fast than our devices [*Nature* **550**, 87–91 (2017)] [*Nat. Electron.* **4**, 681–688 (2021)], and I believe there are also considerable photoconductive gains due to the traps of perovskite films.

But as for the dark current, it directly refers to whether the detector can or cannot be used no matter if it is static or dynamic imaging applications. High dark current can quickly fill up the storage capacitance of TFT or CMOS pixels prior to X-ray illumination, if the capacitance is filled with charges at the dark, there won't be any response from the detector when the X-ray comes in. Thus, compared with the response speed, the high dark current issue is much more urgent to be solved, at least in this early stage of perovskite X-ray detectors

Q2: Second the authors mention that the perovskite photoconductive layer has "low mobility" and the In₂O₃ channel has "superior mobility". For In₂O₃ they measured a mobility of 1.55 cm²/V/s. In the literature they cite (Adv Mater 27, 7168-7175 (2015) 8 cm²/V/s is reported. But the typical mobility values for polycrystalline perovskite reported in lit (0.1-10 cm²/V/s) are in the same range.

A2: We totally understand your concerns. We agree with your opinion about the mobility between In₂O₃ and Perovskite is quite reasonable. Thus, we revised our manuscript that the perovskite has "lower mobility", and the conduction channel, such as In₂O₃, has "higher mobility". In fact, realizing the 3D polycrystalline perovskite with such high carrier mobility as our solution-processed polycrystalline In₂O₃ is still a very challenging work. [*Science Advances*, **7**, 18, (2021)] The major difficulty is the ion migration, which causes a partial screening of the applied field, yielding a very low room temperature μ_{FET} of 10⁻⁴ cm²/Vs in thin films of perovskite, such as MAPbI₃. [*Nat. Commun.* **6**, 7383 (2015).] Even in single-crystal perovskite-based devices, strong hysteresis and moderate mobilities of <10⁻³ cm²/Vs were reported at room temperature. [*Nat. Commun.* **7**, 11330 (2016).] Various instabilities, such as the mechanism of carrier scattering and trapping, the role of ion migration, the origin of hysteresis in the device characteristics, and the electronic structure of grain boundaries in perovskites have made them reveal unusually lower mobility than in the ideal state.

But as for the In₂O₃, their mobility is very high, and stable and is very easy to be reproduced, the calculated electron mobility μ_e [cm²/(V s)] can be up to 270 - 274, the experiment value can be 7.81 - 190. [*Applied Physics Reviews* **9**, 011315 (2022)] They can easily achieve high mobility in various fabrication processes, such as ALD [*J. Phys. Chem. C* **115**, 15384-15389 (2011a)] (84 cm²/(V s)), spin-coating, PLD [*Appl. Phys. Lett.* **62**, 2332-2334 (1993)] (50 cm²/(V s)), MOCVD [*Vacuum* **167**, 1-5 (2019)] (42 cm²/(V s)), spray pyrolysis [*J. Cryst. Growth* **240**, 142-151 (2002)] (42.6 cm²/(V s)), and dc magnetron sputtering. The In₂O₃ film sputtered at room temperature without post-annealing results in layers with reasonably high mobility of 51.3 cm²/(V s). [*Nat. Commun.* **6**, 8932 (2015)]

The In₂O₃ fabricated by spin-coating might have the relatively low mobility, but such a time-saving and low-cost method can be widely used in low-cost large-area applications. What's more, the electron mobility of our In₂O₃ transistor with aluminum Source/Drain electrodes was 8.23 cm²/V·s which is higher than that with Ni/Au Source/Drain electrodes (1.55 cm²/V·s). But unfortunately, the perovskite will have a chemical reaction with aluminum, which may deteriorate the device's stability and performance. We have to use Ni/Au

Source/Drain electrodes as sacrificing the mobility of In_2O_3 to gain stability of the device.

Q3: My third concern is about the thickness. I understand that the novelty here is the device structure and they have not optimized the thickness, interlayers, etc however do they even need 2 microns? The authors mention that the xray generated carriers drift under a built-in electric field between X-ray sensitive material and electron transport layer. In my opinion, the electric field at the ETL interface is not strong and I'm not sure if the depletion region extends 2 microns to the top DSC electrode. I would expect the devices to be limited by charge carrier extraction. There is not much benefit of absorbing more xrays by making thicker films if they cant extract the carriers generated. Moreover, I would expect thicker devices to be worse. Since xray absorption decreases exponentially, for thicker devices most of the carriers generated close to the top surface from where the xray is illuminated might not reach the interface where extraction is more efficient.

A3: Genius perspectives! We appreciate your meticulous reading of our manuscript and your smart views. In perovskite, most of the carriers are indeed generated close to the top surface, if the electric field at the ETL interface is not strong enough to extract the surface photo-induced electrons in perovskite, the thicker geometry might result in poor sensitivity. To figure out this question, we made devices with 2 μm , 1 μm and 500 nm perovskite, the devices were fabricated in the same conditions.

We measured their sensitivities when their DCS electrode is applied with CV, and found that 2 μm device has the highest sensitivity ($7560 \mu\text{Gy}_{\text{air}}^{-1} \text{cm}^{-2}$), the 1 μm device owns a sensitivity of $4060 \mu\text{Gy}_{\text{air}}^{-1} \text{cm}^{-2}$, the 500 nm device reveals the lowest ($2580 \mu\text{Gy}_{\text{air}}^{-1} \text{cm}^{-2}$) (Figure R3). The results seem to point out that in such a relatively thin geometry, the device with thicker perovskite film performs better in collecting photo-induced electrons, the electric field at the ETL interface might strong enough to extract the surface photo-induced electrons in perovskite. Again the relatively high sensitivity should be partially related to the photoconductive gains. Personally, when we talk about the word "extract" we are actually talking about the "drift", but we should also not forget about the "diffusion" especially given the high diffusion length of perovskite crystals. As the drifted charges are collected, it also gives additional driving force for diffusion.

We definitely believe your perspective is generally right, thicker films can generate more photo-induced electrons, thin devices might have better capacity in extracting the majority of the photo-induced electrons at the top of the perovskite. There must be a trade-off between the thickness of perovskite and the device's performance, similar to what Reviewer #1 said, "A trade-off between X-ray absorption, dark current and signal height must be found accurately if industrial applications are envisioned". The 2 μm perovskite in such a device structure might be a relatively thin geometry and the electric field at the ETL interface seems able to extract the photo-induced electrons efficiently. But based on our laboratory experiment condition, we are very sorry we cannot demonstrate the performance of the device with high-quality pure perovskite film thicker than 2 μm (we have shown the thick perovskite-PMMA composite films in the last round revision), because the precursor solution

of perovskite we used is already saturated. We hope the other researchers who have expertise in fabricating thick perovskite films can solve this optimization question in the future. We have added the comments in our revised manuscript (page 12), and thank you very much for your constructive advice.

Figure R3. Sensitivity of devices with different thicknesses.

Other minor comments:

Q4: Line 128 and 133 CV has been defined earlier. The authors could avoid using acronyms for terms that have not been used many times such as WV, PCD. They are not commonly used acronyms and makes it difficult to read.

A4: Thank you very much for your careful review, we have deleted the WV and PCD in the manuscript, and added an annotation in the caption.

Q5: Line 172 "Our PMMA layer is only ~50 nm, it cannot act as a dielectric layer."

Thinner layers have higher capacitance than thicker layer.

A5: Thank you very much for your notification. The PMMA layer is not insulative but may introduce a capacitance between the DCS electrode and perovskite. From the device's output signal, the capacitance seems will not affect the final signal, which is collected in the Drain electrode on the other side of the perovskite. We have added some comments on the capacitance of the PMMA in our manuscript (page 9).

Q6: Line 216 "The detection limit of the DCS detector 217 (SNR = 3) is as low as 7.84 nGyair s⁻¹(Fig. 4g),"

The authors should mention how they obtained this number in the manuscript (used reverse extension line to estimate the lowest detection....?)

A6: Thank you for your suggestions. We have added a comment "The lowest detection limit was estimated by the reverse extension line to where the SNR = 3" in the revised manuscript.

(page 15)

Q7: Line 242 "To preliminarily investigate its spatial resolvability in scanning-based X-ray imaging, we measured its modulation transfer function(MTF) by scanning the object (Figure S11a). The used line pair mask plate is shown in Figure S11b, the minimum resolved line pair is 5.5 lp mm⁻¹."

Could the authors show the scanned x-ray image of the object and include it in SI? Was this a single pixel device? MTF for the detector array would be more relevant. Did they measure it?

A7: Thank you for your suggestions. Yes, we calculated the MTF with a single-pixel device, we preliminarily calculated the MTF by scanning the object with a stepping motor, and recording the current value of a single-pixel device. We added this comment in our revised manuscript to avoid misleading the readers (page 12, 13). Indeed, the MTF was sometimes calculated by scanning the whole X-ray image of the object, but as you can see, every scanning line to this object is the same, when calculating the MTF, only need to scan one of the lines can preliminarily calculate the MTF. As for MTF for our imaging array, we are really sorry that we didn't measure it. Even the world's most state-of-the-art perovskite imaging array fabricated by Samsung (1,428 × 1,428 pixels in 10 cm × 10 cm) with a pixel pitch of 70 μm) [*Nature* **550**, 87–91 (2017)], their MTF for the detector array was just 3.1 lp mm⁻¹. In our preliminarily demonstrated detector array, there are only 64 × 64 pixels in 2cm × 2cm with a pixel pitch of 300 μm. Thus, MTF for our detector array can be negligible. We aim to demonstrate a prototype to reveal that this method can be successfully used in an imaging array and hope to promote the method rather than the detailed performance factors of the array.

Q8: Line 255 "of dark current drafting" Typo

A8: Thank you for your notification. We have revised the "drafting" to "drifting" in Line 255.

Q9: Figure still have a lot of text in small font. They could put only the most relevant/important text in the figure and the description could go in the caption or in the main text.

For eg, they could remove or reduce in fig 2. direction of dark electron motion, photoconductive gain. In fig 4 almost no shift of dark current drifting

Fig 3a text too small

A9: Thank you for your advice. We have improved the small font and deleted some irrelevant texts in the figure.